# Trajectory Tracking and Adaptive Fuzzy Vibration Control of Multilink Space Manipulators with Experimental Validation

**Chenlu Feng, Weidong Chen, Minqiang Shao * and Shihao Ni**

State Key Laboratory of Mechanics and Control for Aerospace Structures, Nanjing University of Aeronautics and Astronautics, Nanjing 210016, China
* Correspondence: m.q.shao@nuaa.edu.cn

**Abstract:** This paper investigates the problem of modeling and controlling a space manipulator system with flexible joints and links. The dynamic model of the flexible manipulator system is derived by using the Lagrange equation and the floating frame of reference formulation, where the assumed mode method is adopted to discretize flexible links, while the flexible joints are regarded as linear torsion springs. The natural characteristics of a single flexible link manipulator, under three different boundary conditions, are compared to reveal the effect of the flexibility of joints on the manipulator system and to choose suitable assumed modes. Furthermore, singular perturbation theory is introduced to decompose the system into a slow subsystem that describes the rigid-body motion, and a fast subsystem that describes the elastic vibration. Since the system is underactuated, a compound control strategy, which consists of the underactuated computed torque controller and the adaptive fuzzy controller, is presented to improve the accuracy of the trajectory tracking of the flexible joints and to suppress the elastic vibration of the flexible links, in the meantime. Both numerical simulation and experimentation are performed to verify the effectiveness of the proposed compound controller, and a comparison with the proportional-derivative (PD) controller is provided to highlight its superiority in suppressing the residual vibration of the tip.

**Keywords:** flexible space manipulator; flexible joint; singular perturbation theory; underactuated computed torque controller; adaptive fuzzy controller

## 1. Introduction

Space manipulators play an important role in on-orbit activities, such as construction, inspection, and transportation [1,2]. With the development of space technology, lighter and larger space manipulators are increasingly applied because of their advantages of being lightweight, having low energy consumption, quick responses [3]. However, there exist obvious flexible characteristics in the space manipulator. They are mainly caused by the structural flexibility of the links and the flexibility of the joints with harmonic gear reducers. The elastic vibrations generated by these two kinds of flexibility are highly coupled, which complicates the dynamic characteristics of the space manipulator system [4] and puts forward greater requirements for controlling. In addition, the coupling between the free-floating base and the manipulator brings challenges to the control of the manipulator [5]. Structural vibrations will be obvious when the flexible manipulators perform on-orbit missions, especially for large space flexible manipulators, which may lead to a catastrophic failure. On the other hand, flexible manipulators usually have high dimensional orders, low damping ratios, and parameter uncertainties in dynamics. Therefore, it is necessary and challenging work to investigate the control strategy of the space flexible manipulator.

The focus of controlling a flexible manipulator is to track the desired trajectories and suppress the vibration of the flexible parts. The dynamics and control of the manipulators have been studied for a long time. Based on differences in modeling, control, and experimental studies, Dwivedy and Eberhard [6] summarized the original works in the

field of the dynamic performance of flexible robots. In recent years, many researchers have extensively studied the problem of modeling and planning and paid more attention to the controlling of the flexible space manipulator, which has been applied to different robot platforms [7–9]. At present, advanced intelligent materials have been applied in the research of some active control methods [10–12].

A high-precision mathematical model is crucial for controller design. However, a high-precision mathematical model is often difficult to be obtained due to the uncertainty and error in the model. Yang [13] creatively introduced a set of filtered error variables and asymptotic filters as well as an auxiliary system, while two novel continuous integral robust control algorithms have been synthesized, via an improved backstepping framework, for a class of high-order systems suffering from both matched and mismatched disturbances. The validation of the proposed controller is performed on a single-link rigid manipulator and a two-link rigid manipulator, respectively. Shawky [14] used a nonlinear controller via the state-dependent Riccati equation (SDRE) to compensate for the uncertainties of the single-link flexible manipulator system with a rigid joint. The simulation results verified the effectiveness of the SDRE controller. De Luca [15] considered two model classes: robots with elastic joints and rigid links, and robots with flexible links and rigid joints. In view of the small deformation, the elasticity of the elastic joint was modeled as a linear spring. Then, model-based feedforward laws were derived for two basic motion tasks, although the generalization of the control scheme to a multilink flexible arm was a problem. The intelligent control technology, which does not depend on models, has paid more attention to the suppression of the vibration of the flexible links. Qiu et al. [16] introduced a hybrid control strategy of optimal trajectory planning and diagonal recurrent neural network (DRNN) control to suppress the vibration of a single-link flexible manipulator with a rigid joint, both during and after the point-to-point motion. Experimental results demonstrated that planning an optimal trajectory could cause fewer vibrations and that the DRNN controller was superior to the classical PD controller on vibration suppression. Malzahn et al. [17] presented a conjunction of a model-free independent joint control strategy for vibration damping with a neural-network-based payload estimation and an inverse kinematics model based on multilayer perceptron (MLP) networks for a multilink-flexible robot arm under gravity, the control architecture enabled the robot arm to catch multiple balls sequentially thrown by a human. Neural networks have been used in controlling flexible manipulators because of their strong nonlinear fitting ability, yet training requires a lot of available training data, and the training time is often long. Cao et al. [18] developed a fuzzy self-tuning proportional-integral-derivative (PID) controller and applied it to a two-link flexible manipulator featuring a piezoelectric ceramics (PZT) actuator, and the experimental results showed that the controller could effectively suppress the vibration. Owing to the existence of nonlinear factors and parameter uncertainties, model-based control methods cannot maintain the required accuracy. Wei et al. [19] designed model-free fuzzy logic control laws to suppress the vibrations of the single-link flexible piezoelectric manipulator. Experimental results showed that the adopted fuzzy control algorithms could substantially suppress the larger amplitude vibrations. Qiu et al. [20] utilized a Takagi–Sugeno model-based fuzzy neural network control (TS-FNN) scheme to suppress the residual vibrations of the two-link flexible manipulator with rigid joints. Experimental results demonstrated that the designed controller could reduce the residual vibrations quicker than the traditional linear PD controller. Tracking the desired trajectory of a flexible joint is the focus of controlling flexible manipulators. In order to achieve high-precision tracking of the revolving angles and vibration suppression of the elastic part, Zhang et al. [21] developed an adaptive iterative learning control (AILC) law for a two-link rigid–flexible coupled manipulator system with rigid joints in a three-dimensional (3D) space. The computed torque method (CTM) has been maturely applied to the tracking of the joint. Mehrzad et al. [22] designed a modified CTM to control the manipulator motion. Using numerical simulations, the performance of the proposed control system was evaluated for

trajectory tracking. In addition, the assumed modes method was mostly used for modeling flexible links in the aforementioned references.

At present, most research paid more attention to rigid–flexible coupled space manipulator systems with a rigid joint. However, the flexibilities of the joints and links exist objectively only in practice. Thus, there is limited research on the types of assumed modes when using the assumed modes method to model flexible manipulators. Most controllers have high requirements for model accuracy. In practice, frictions, interstices, and impacts in gear transmission of manipulator systems are hard to model, and accurate mathematical models are hard to be obtained. In addition, according to the singular perturbation model of the space manipulator system, the fast-varying subsystem featured by the flexible vibration has the form of a linear equation. Although many controllers exist for linear systems, such as the linear quadratic regulator (LQR) controllers [23], state feedback control, etc., their performance greatly relies on the accuracy of the mathematical model of the controlled system. Therefore, it is necessary to investigate model-independent controllers. A fuzzy control does not require an accurate mathematical model and has the characteristic of good anti-interference. Moreover, an adaptive fuzzy controller (AFC) combines the advantages of traditional fuzzy control and an adaptive learning algorithm and, thus, can cope with the modeling error and the external disturbance excitation in the motion. Hence, it is a more desirable choice to design the AFC for the reduced flexible vibration of the system.

The objectives of this paper are to improve the accuracy of trajectory tracking for the flexible joints and suppress the elastic vibrations of the flexible links by using a compound control strategy that consists of the underactuated computed torque controller and the adaptive fuzzy controller. Simulations and experiments, then, confirmed the effectiveness of the proposed controller. A conclusion was drawn based on the reported results. The main contributions of this paper are as follows:

1. A mathematical model of a multilink flexible space manipulator system with flexible joints and links was established.
2. The dynamic responses and natural characteristics of the flexible manipulator under three different kinds of mode shapes are compared.
3. Based on the underactuated characteristic of the system, the underactuated CTM is designed to achieve high-precision performance on flexible joint trajectory tracking, and the non-model adaptive fuzzy controller is adopted to suppress the elastic vibrations of the flexible links.

## 2. Dynamics of Flexible Space Manipulator System

### 2.1. Mathematical Model of Flexible Space Manipulator System

A common space flexible manipulator system is shown in Figure 1. It consists of flexible links with a uniform cross-sectional area, flexible joints, and a free-floating base. A pair of PZT actuators are attached to the root of each flexible link [19] and a tip payload is attached to the distal end of the last flexible link. The entire system rotates in the horizontal plane, driven by electric motors. The kinematic and dynamic symbols of the space manipulator system used in this paper are listed in Table 1. Unless otherwise specified, all reference frame systems are inertial frames. In Figure 1, $O_0$ is the position of the center of mass (CM) of the free-floating base, $O_i$ $(i = 1, \ldots, n)$ is the position of the $i$th joint's CM, $\mathbf{p}_i \in \mathbf{R}^2$ $(i = 1, \ldots, n)$ is the position vector of the $i$th joint's CM in the inertial frame $\Sigma_d$, $\mathbf{r}'_i \in \mathbf{R}^2$ $(i = 1, \ldots, n)$ is the position vector of a point $P$ on the $i$th link in the frame $\hat{\Sigma}_i$, and $\mathbf{r}_i \in \mathbf{R}^2$ $(i = 1, \ldots, n)$ is the position vector of a point on the $i$th link in the inertial frame $\Sigma_d$, respectively.

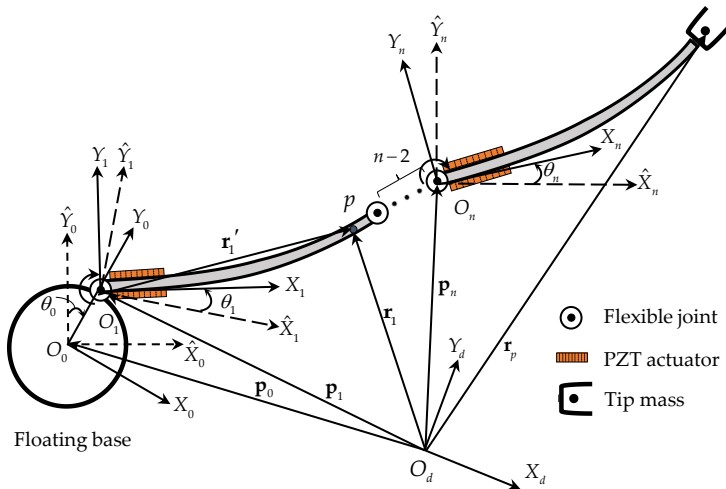

**Figure 1.** Schematic of a flexible manipulator system.

**Table 1.** Kinematic and dynamic symbols of the space manipulator system.

| Symbol | Representation |
| --- | --- |
| $J_0, J_i(i = 1, \ldots, n)$ | Moment of inertia of the base, the $i$th rotor |
| $m_0, m_i(i = 1, \ldots, n)$ | mass of the base, the $i$th rotor |
| $l_0, l_1(i = 1, \ldots, n)$ | length of the base, the $i$th link |
| $\rho_i(i = 1, \ldots, n)$ | The linear density of the $i$th link |
| $I_i(i = 1, \ldots, n)$ | The moment of inertia of the $i$th link |
| $E_i(i = 1, \ldots, n)$ | The elastic modulus of the $i$th link |
| $m_p$ | The mass of the tip payload |
| $J_p$ | Moment of inertia of tip payload |
| $k_i(i = 1, \ldots, n)$ | The $i$th spring rate coefficient |
| $\alpha_i(i = 1, \ldots, n)$ | The theoretical rotation angle of the $i$th rotor |
| $\theta_i(i = 1, \ldots, n)$ | The actual rotation angle of the $i$th joint |
| $\tau_i(i = 1, \ldots, n)$ | The theoretical torque of the $i$th joint |
| $\sigma_i(i = 1, \ldots, n)$ | The elastic deformation of the $i$th joint |
| $\Sigma_0, \Sigma_i(i = 1, \ldots, n)$ | Base, link frame system |
| $\hat{\Sigma}_i(i = 1, \ldots, n)$ | Joint frame system |
| $\Sigma_d$ | Inertial frame system |

The flexibility of the space manipulator comes from the flexibility of the joints and the structural flexibility of the links. In practice, the joint deformations are small, and, thus, the elasticity in the joints can be modeled as a spring [15]. All electric motors were assumed as uniform rotors with their centers of mass on the rotation axes [24]. Figure 2 shows the revolute joint model established in this paper. According to geometry, one has:

$$\sigma_i = \alpha_i - \theta_i. \tag{1}$$

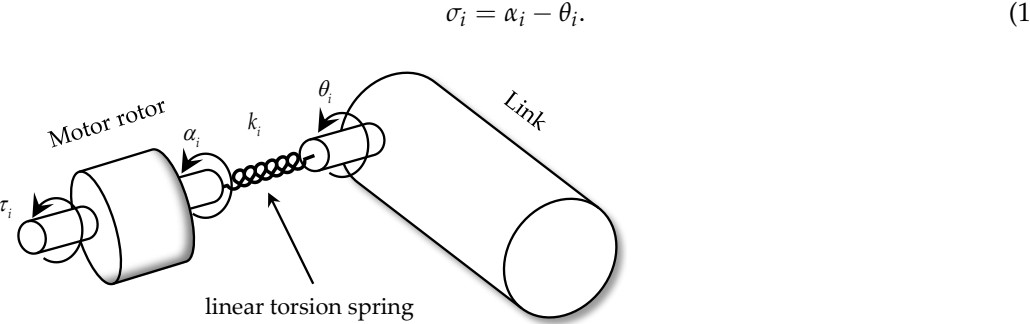

**Figure 2.** Schematic of the joint model.

Thus, by using the assumed mode method and the assumption of small deformations, the *ith* link's bending deformation can be expressed in terms of $m$ mode shapes as:

$$u_i(x,t) = \sum_{s=1}^{m} q_i^s(t)\varphi_s^i(x),$$ (2)

where $\varphi_s^i$ and $q_i^s$ denote the *ith* modal shape and the corresponding generalized coordinate, respectively.

Thus, the position vector $\mathbf{r}'_i \in \mathbf{R}^2$ is:

$$\mathbf{r}'_i = \begin{bmatrix} x \\ u_i(x,t) \end{bmatrix}.$$ (3)

The torque of the *jth* PZT attached to the *ith* link can be expressed as $M_i^j = K_a U_j$, where $K_a$ is a constant related to the natural characteristics of piezoelectric ceramics.

By using the floating frame of the reference formulation, the motion of the flexible manipulator system was regarded as the superposition of the large-scale rigid body motion and the deformation of flexible links. Hence, the manipulator system comprises $n + 1$ bodies and $n$ hinges. The three kinds of frames are listed in Table 1.

In order to derive the mathematical model of the manipulator system, the matrix $\mathbf{S}_i$ of converting the floating frame to the inertial frame should be given. In Figure 3, the angle transformed from frame $\Sigma_i$ to frame $\hat{\Sigma}_i$ is the sum of the flexible rotation angle $\phi_{i-1}|_{x=l_{i-1}}$ of $(i-1)th$ link's end and the *ith* joint's rotation angle $\theta_i$. Thus, the matrix of converting the floating frame to the inertial frame is:

$$\mathbf{S}_i = \mathbf{S}_{i-1}\mathbf{E}_i\mathbf{A}_i = \hat{\mathbf{S}}_{i-1}\mathbf{A}_i, \ \hat{\mathbf{S}}_0 = \mathbf{I}_{2\times2},$$ (4)

where $\mathbf{A}_i = \begin{bmatrix} \cos(\theta_i) & -\sin(\theta_i) \\ \sin(\theta_i) & \cos(\theta_i) \end{bmatrix}$, $\mathbf{E}_i = \begin{bmatrix} \cos(\phi_{i-1}) & -\sin(\phi_{i-1}) \\ \sin(\phi_{i-1}) & \cos(\phi_{i-1}) \end{bmatrix}\Big|_{x=l_{i-1}}$, $\mathbf{E}_1 = \mathbf{I}_{2\times2}$.

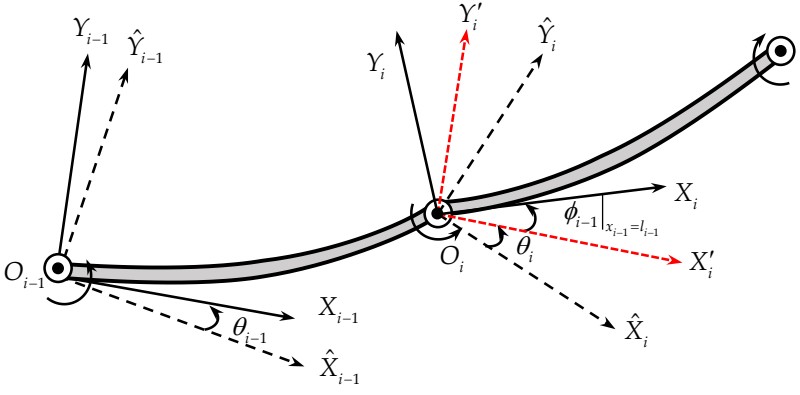

**Figure 3.** Schematic of two adjacent links in a manipulator system.

Based on the assumption of a small deformation, the matrix $\mathbf{E}_i$ can be simplified to:

$$\mathbf{E}_i = \begin{bmatrix} 1 & -\phi_{i-1} \\ \phi_{i-1} & 1 \end{bmatrix}\Big|_{x=l_{i-1}}.$$ (5)

Therefore, the position vector $\mathbf{r}_i(x)$ in the inertial frame can be written as:

$$\mathbf{r}_i(x) = \mathbf{S}_i\mathbf{r}'_i + \mathbf{p}_i, \ \mathbf{p}_{i+1} = \mathbf{p}_i + \mathbf{S}_i\mathbf{r}'_i(l) = \mathbf{r}_i(l).$$ (6)

The velocity of $\mathbf{r}_i(x)$ on the *ith* flexible link is:

$$\dot{\mathbf{r}}_i(x) = \dot{\mathbf{S}}_i \mathbf{r}_i' + \mathbf{S}_i \dot{\mathbf{r}}_i' + \dot{\mathbf{p}}_i, \tag{7}$$

where $\dot{\mathbf{S}}_i = \dot{\hat{\mathbf{S}}}_{i-1} \mathbf{A}_i + \hat{\mathbf{S}}_{i-1} \dot{\mathbf{A}}_i$, $\dot{\hat{\mathbf{S}}}_i = \dot{\mathbf{S}}_i \mathbf{E}_{i+1} + \mathbf{S}_i \dot{\mathbf{E}}_{i+1}$, $\dot{\mathbf{A}}_i = \mathbf{B} \mathbf{A}_i \dot{\theta}_i$, $\dot{\mathbf{E}}_i = \mathbf{B} \dot{\phi}_i \big|_{x=l}$, $\mathbf{B} = \begin{bmatrix} 0 & -1 \\ 1 & 0 \end{bmatrix}$.

To derive the equation of motion for the space manipulator, which consists of lumped parameter parts (the joint rotations) and distributed parameter parts (the link deformations), by using the Lagrange formulation, the kinetic energy $T$ and the potential energy $V$ of the system must be computed. The kinetic energy $T$ receives contributions from the base, links, joints, and tip payload and is given by:

$$T = T_0 + T_l + T_\alpha + T_p, \tag{8}$$

where

$$T_0 = \frac{1}{2}(m_0 \dot{\mathbf{r}}_0^T \dot{\mathbf{r}}_0 + J_0 \dot{\varphi}_0^2), \tag{9}$$

$$T_l = \frac{1}{2}\sum_{i=1}^n \int_0^{l_i} \rho_i \dot{\mathbf{r}}_i^T \dot{\mathbf{r}}_i dx, \tag{10}$$

$$T_\alpha = \frac{1}{2}\sum_{i=1}^n (J_i \dot{\alpha}_i^2 + m_i \dot{\mathbf{r}}_i^T(0) \dot{\mathbf{r}}_i(0)), \tag{11}$$

$$T_P = \frac{1}{2}m_P \dot{\mathbf{r}}_n^T(l) \dot{\mathbf{r}}_n(l) + \frac{1}{2}J_p(\dot{\varphi}_{ne})^2. \tag{12}$$

In Equation (12), $\dot{\varphi}_{ne} = \sum_{j=1}^n \dot{\theta}_j + \sum_{k=1}^n \dot{\phi}_k \big|_{x=l_k}$.

The links of the space manipulator are modeled as Euler–Bernoulli beams. Since the space manipulator system is in a weightless environment, the effect of gravity can be ignored. Therefore, the potential energy $V$ is only contributed to by the links and joints, i.e.,

$$V = V_l + V_\alpha, \tag{13}$$

where

$$V_\alpha = \frac{1}{2}\sum_{i=1}^n k_i(\alpha_i - \theta_i)^2, \tag{14}$$

$$V_l = \frac{1}{2}\sum_{i=1}^n \int_0^{l_i} \left[ EI_i\left(\frac{\partial \phi_i}{\partial x}\right)^2 \right] dx. \tag{15}$$

The generalized Lagrange Equation [14] of the second kind is:

$$\frac{d}{dt}\left(\frac{\partial T}{\partial \dot{q}_j}\right) - \frac{\partial T}{\partial q_j} + \frac{\partial V}{\partial q_j} = f_j. \tag{16}$$

The system dynamic equations can be obtained by substituting Equations (8) and (13) into Equation (16) and is given by:

$$\begin{cases} \mathbf{J}\ddot{\alpha} + \mathbf{K}_\sigma(\alpha - \theta_{sub}) = \tau_{n \times 1}, \\ \mathbf{M}(\theta, \mathbf{q})\begin{bmatrix} \ddot{\mathbf{x}}_0 \\ \ddot{\theta} \\ \ddot{\mathbf{q}} \end{bmatrix} + \mathbf{F}(\theta, \dot{\theta}, \mathbf{q}, \dot{\mathbf{q}}) + \begin{bmatrix} \mathbf{0}_{3 \times 1} \\ -\mathbf{K}_\sigma(\alpha - \theta_{sub}) \\ \mathbf{K}_q \mathbf{q} \end{bmatrix} = \begin{bmatrix} \mathbf{0}_{(n+3) \times 1} \\ \tau'_{mn \times 1} \end{bmatrix}, \end{cases} \tag{17}$$

where $\boldsymbol{\theta} = [\theta_0, \theta_1, \cdots, \theta_n]^T$, $\boldsymbol{\alpha} = [\alpha_1, \alpha_2, \cdots, \alpha_n]^T$, $\boldsymbol{\tau}_{n \times 1} = [\tau_1, \tau_2, \cdots, \tau_n]^T$ is the torque vector of the joint motor. Moreover, $\mathbf{q} = [q_1^1, q_1^2, \cdots, q_1^m, \cdots, q_i^1, q_i^2, \ldots, q_i^m, \cdots q_n^m]^T$ is a vector consisting of flexible links modal coordinates, $\boldsymbol{\theta}_{sub} = [\theta_1, \cdots, \theta_n]^T$, $\mathbf{x}_0 = [x_0, y_0]^T$ is the position vector of the free-floating base, $\mathbf{M}(\boldsymbol{\theta}, \mathbf{q})$ is the symmetric inertial matrix, $\mathbf{F}(\boldsymbol{\theta}, \dot{\boldsymbol{\theta}}, \mathbf{q}, \dot{\mathbf{q}})$ is the coupled term characterizing the interactions between centrifugal force and Coriolis force, $\mathbf{J} = diag(J_1, \ldots, J_n)$, $\mathbf{K}_\sigma = diag(k_1, \ldots, k_n)$, and $\mathbf{K}_q$ are the joint rotors mass matrix, joints stiffness matrix, and links stiffness matrix, respectively. $\mathbf{K}_q$ is expressed as:

$$\mathbf{K}_q = \begin{bmatrix} \int_0^{l_1} EI(\frac{d^2\boldsymbol{\Phi}^1}{dx^2}) \times (\frac{d^2\boldsymbol{\Phi}^1}{dx^2})^T dx & & \\ & \ddots & \\ & & \int_0^{l_n} EI(\frac{d^2\boldsymbol{\Phi}^n}{dx^2}) \times (\frac{d^2\boldsymbol{\Phi}^n}{dx^2})^T dx \end{bmatrix}, \tag{18}$$

where $\boldsymbol{\Phi}^i = [\varphi_1^i, \ldots, \varphi_m^i]^T$ is a vector consisting of the *ith* link modal shape functions and $\boldsymbol{\tau}'_{mn \times 1}$ is the torque vector of PZT. The generalized force of the *jth* PZT attached to the *ith* link can be expressed as:

$$\tau_i^j = \int_{b_{ij}}^{b_{ij}+s_{ij}} \sum_{v=1}^m M_i^j \frac{d\varphi_v^i(x)}{dx} = \sum_{v=1}^m M_j^i (\frac{d\varphi_v^i(b_{ij}+s_{ij})}{dx} - \frac{d\varphi_v^i(b_{ij})}{dx}), \tag{19}$$

where $b_{ij}$ and $s_{ij}$ are the start- and endpoints of the *jth* PZT attached to the *ith* link, respectively.

The torque vector $\boldsymbol{\tau}'_{mn \times 1}$ can be expressed as:

$$\boldsymbol{\tau}'_{mn \times 1} = \mathbf{P}_{nm \times nm} \times \mathbf{M}_{nm \times 1} = \begin{bmatrix} \mathbf{P}_1 & & \\ & \ddots & \\ & & \mathbf{P}_n \end{bmatrix} \times \begin{bmatrix} \mathbf{M}_1 \\ \vdots \\ \mathbf{M}_n \end{bmatrix}, \tag{20}$$

where $\mathbf{M}_i = [M_1 \cdots M_m]^T$, the matrix $\mathbf{P}_i$ can be expressed as:

$$\mathbf{P}_i = \begin{bmatrix} \frac{d\varphi_1^i(b_{i1}+s_{i1})}{dx} - \frac{d\varphi_1^i(b_{i1})}{dx} & \cdots & \frac{d\varphi_1^i(b_{im}+s_{im})}{dx} - \frac{d\varphi_1^i(b_{im})}{dx} \\ \vdots & \ddots & \vdots \\ \frac{d\varphi_m^i(b_{i1}+s_{i1})}{dx} - \frac{d\varphi_m^i(b_{i1})}{dx} & \cdots & \frac{d\varphi_m^i(b_{im}+s_{im})}{dx} - \frac{d\varphi_m^i(b_{im})}{dx} \end{bmatrix}. \tag{21}$$

Substituting $M_i^j = K_a U_j$ into Equation (20) yields:

$$\boldsymbol{\tau}'_{mn \times 1} = \mathbf{P} \times \begin{bmatrix} \mathbf{M}_1 \\ \vdots \\ \mathbf{M}_n \end{bmatrix} = \mathbf{P} \times \begin{bmatrix} \mathbf{K}_1 & & \\ & \ddots & \\ & & \mathbf{K}_n \end{bmatrix} \times \begin{bmatrix} \mathbf{U}_1 \\ \vdots \\ \mathbf{U}_n \end{bmatrix} = \mathbf{K}_v \mathbf{U}, \tag{22}$$

where $\mathbf{K}_1 = \begin{bmatrix} K_a & & \\ & \ddots & \\ & & K_a \end{bmatrix}_{m \times m}$.

The first and the final equations of the dynamic model (17) are referred to as the motor and link equations, respectively.

## 2.2. Natural Characteristics of the Flexible Links under Different Boundary Conditions

Employing the assumed mode method, one can easily obtain the approximated link deformation by using only a finite number of modes. However, the model accuracy is not only affected by the number of modes but also by the kind of selected modes. Many researchers have investigated the errors introduced by modal truncation, and the cantilever beam mode and the simply supported beam mode are commonly used [25,26]. In fact, the

boundary conditions of the flexible links of the space manipulators are different from those of cantilever beams and simply supported beams; thus, using either one of the two modes may introduce errors. Considering the actual boundary condition of the space manipulator, the natural characteristics of the flexible links, under three different boundary conditions, are compared with each other to investigate the influence of the flexibility of the joints. The three boundary conditions are:

1. Fixed-free boundary condition, which is a single cantilever beam (SCB) boundary condition.
2. Fixed-inertial load boundary condition, which is a single rigid joint and flexible link manipulator (SRF) boundary condition.
3. Elastic load-inertial load boundary condition, which is a single flexible joint and flexible link manipulator (SFF) boundary condition.

As mentioned previously, the link is modeled as the Euler–Bernoulli beam. The free vibration differential equation of the Euler–Bernoulli beam [10] is:

$$\rho\frac{\partial^2 u}{\partial t^2} + EI\frac{\partial^4 u}{\partial x^4} = 0, \tag{23}$$

where $\rho$ is mass per unit length and $EI$ is the bending stiffness.

The dimensionless parameters are defined as:

$$\xi = \frac{x}{l}, \Gamma_p = \frac{J_p}{\rho l^3}, M_p = \frac{m_p}{\rho l}, s = \left(\frac{\rho l^4 \omega^2}{EI}\right)^{\frac{1}{4}}. \tag{24}$$

The ratio of the stiffness of the flexible joint to the bending stiffness of the flexible link is defined as:

$$k_m = \frac{kl}{EI}. \tag{25}$$

The general solution of Equation (23) can be expressed as:

$$u(x,t) = \varphi(x)\sin(\omega t) = l\varphi(\xi)\sin(\omega t). \tag{26}$$

Substituting Equation (26) into Equation (23) yields a dimensionless expression:

$$\varphi(\xi)^{(4)} - s^4\varphi(\xi) = 0. \tag{27}$$

The general solution of Equation (27) can be expressed as:

$$\varphi(\xi) = A_1 \cdot \cos(s \cdot \xi) + A_2 \cdot \sin(s \cdot \xi) + A_3 \cdot \cosh(s \cdot \xi) + A_4 \cdot \sinh(s \cdot \xi). \tag{28}$$

The dimensionless boundary condition formulas corresponding to the three boundary conditions are:

1. SCB boundary condition:

$$\varphi(0) = 0, \left.\frac{\partial\varphi(\xi)}{\partial\xi}\right|_{\xi=0} = 0, \left.\frac{\partial^2\varphi}{\partial\xi^2}\right|_{\xi=l} = 0, \left.\frac{\partial^3\varphi}{\partial\xi^3}\right|_{\xi=l} = 0. \tag{29}$$

2. SRF boundary condition:

$$\varphi(0) = 0, \left.\frac{\partial\varphi(\xi)}{\partial\xi}\right|_{\xi=0} = 0, \left.\frac{\partial^2\varphi}{\partial\xi^2}\right|_{\xi=1} - s^4\Gamma_p\left(\left.\frac{\partial\varphi}{\partial\xi}\right|_{\xi=1}\right) = 0, \left.\frac{\partial^3\varphi}{\partial\xi^3}\right|_{\xi=1} + s^4 M_p\frac{d^2}{dt^2}\left(\varphi|_{\xi=1}\right) = 0. \tag{30}$$

3. SFF boundary condition:

$$
\begin{aligned}
&\varphi(0) = 0, \\
&\left.\frac{\partial^2 \varphi(\xi)}{\partial \xi^2}\right|_{\xi=0} - k_m \left.\frac{\partial \varphi(\xi)}{\partial \xi}\right|_{\xi=0} = 0, \\
&\left.\frac{\partial^2 \varphi}{\partial \xi^2}\right|_{\xi=1} - s^4 \Gamma_p \left(\left.\frac{\partial \varphi}{\partial \xi}\right|_{\xi=1}\right) = 0, \\
&\left.\frac{\partial^3 \varphi}{\partial \xi^3}\right|_{\xi=1} + s^4 M_p \frac{d^2}{dt^2}\left(\varphi|_{\xi=1}\right) = 0.
\end{aligned}
\tag{31}
$$

Substituting the general solution (28) into the aforementioned boundary condition formulas, respectively, provides the three corresponding frequency equations:

1. SCB boundary condition:

$$
\cosh(s)^2 + 2\cosh(s)\cos(s) - \sinh(s)^2 + \sin(s)^2 + \cos(s)^2 = 0. \tag{32}
$$

2. SRF boundary condition:

$$
\begin{aligned}
&-2s\sinh(s)\left(s^2\Gamma_p - M_p\right)\cos(s) + 2s^4\Gamma_p M_p + 2 \\
&+ \left(\left(-2s^4\Gamma_p M_p + 2\right)\cos(s) - 2s\sin(s)\left(s^2\Gamma_p + M_p\right)\right)\cosh(s) = 0
\end{aligned}
\tag{33}
$$

3. SFF boundary condition:

$$
\begin{aligned}
&2s\sinh(s)\left(s^4\Gamma_p M_p + s^2\Gamma_p k_m - M_p k_m - 1\right)\cos(s) \\
&- 2s^4\Gamma_p M_p k_m + 4\sinh(s)\sin(s)s^2 M_p - 2k_m \\
&+ \left[\left(\left(2\Gamma_p M_p k_m + 4\Gamma_p\right)s^4 - 2k_m\right)\cos(s) - \right. \\
&\left. 2s\sin(s)\left(s^4\Gamma_p M_p - s^2\Gamma_p k_m - M_p k_m - 1\right)\right]\cosh(s) = 0
\end{aligned}
\tag{34}
$$

The parameters $M_p$, $k_m$, and $\Gamma_p$ directly affect the natural characteristics of the manipulator system, thus, the effect of the parameters on the natural characteristics is worth studying in detail. According to the obtained frequency Equations (32)–(34), the relationship curve between the dimensionless parameters and the dimensionless natural frequencies can be drawn.

In many missions, space manipulators only need to grasp light objects, in which case the dead-weight load ratio $M_p$ is small. Setting $M_p = 0.1$, the 3D surfaces of the first two dimensionless frequencies $s_1$ and $s_2$ versus the stiffness ratio $k_m$ and the moment of inertia $\Gamma_p$ are shown in Figures 4 and 5, respectively. The dimensionless frequency surfaces of SRF are below the dimensionless frequency surfaces of SFF, as $\Gamma_p$ increases, the first-order dimensionless frequency surface of SRF approaches the first-order dimensionless frequency surface of SFF, while the second-order dimensionless frequency surfaces are far away from each other, as shown in Figures 4 and 5, respectively. With the increase in the stiffness ratio $k_m$, the first two dimensionless frequency surfaces of SRF or SFF gradually move away from those of SCB, respectively, while the first two dimensionless frequency surfaces of SFF gradually approach the surfaces of SRF. It can be concluded that $\Gamma_p$ and $k_m$ have different influences on each order of the dimensionless frequency surface.

Grabbing and releasing payloads are important for space manipulators to perform on-orbit missions. Therefore, it is necessary to analyze the effect of the load ratio $M_p$ on the natural characteristics of the flexible manipulators. When $k_m = 0.5$, the 3D surfaces of the first two dimensionless frequencies $s_1$ and $s_2$ versus $M_p$ and $\Gamma_p$ are shown in Figures 6 and 7, respectively. Following the increase of the parameter $M_p$, the first-order dimensionless frequency surface of SFF is far away from that of SRF, while the second-order dimensionless frequency surfaces approach each other, as shown in Figures 6 and 7, respectively. Therefore, an increase in both $M_p$ and $\Gamma_p$ can reduce the dimensionless frequencies of SRF and SFF, although the effect of $M_p$ and $\Gamma_p$ on the differences in the dimensionless frequencies of the same order of SRF and SFF is the opposite.

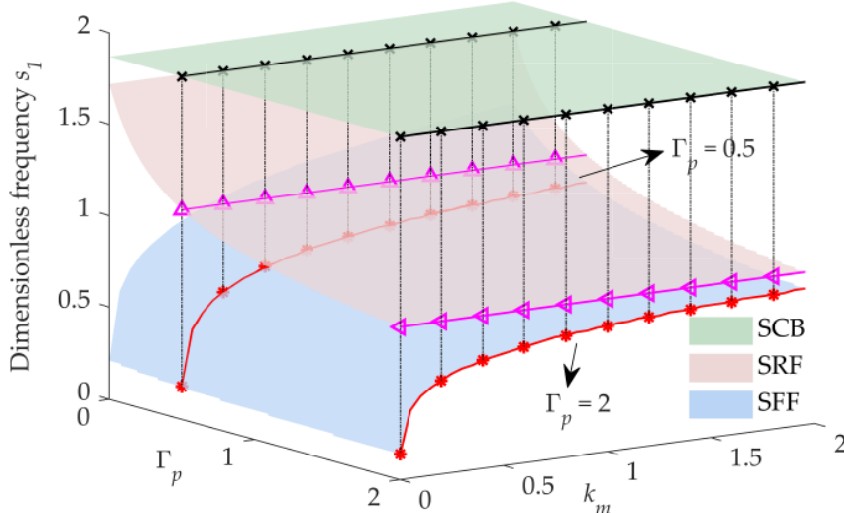

**Figure 4.** 3D surfaces of the first frequency versus $k_m$ and $\Gamma_p$ ($M_p = 0.1$).

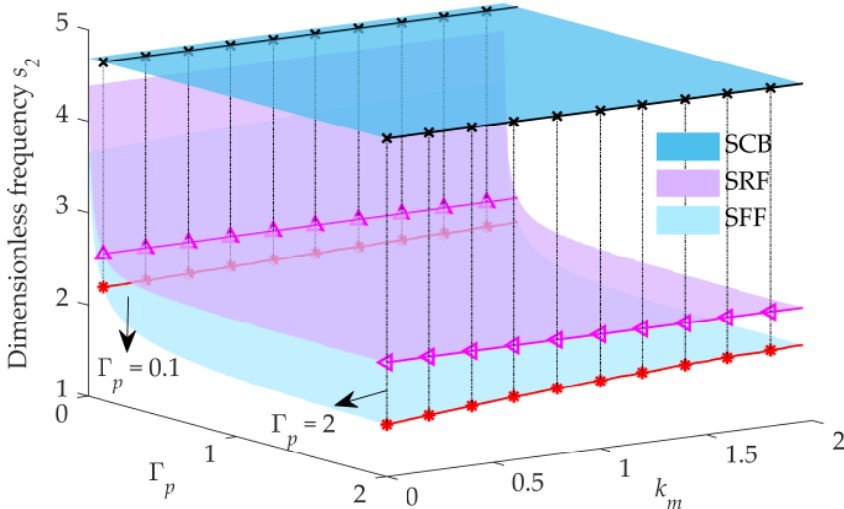

**Figure 5.** 3D surfaces of the second frequency versus $k_m$ and $\Gamma_p$ ($M_p = 0.1$).

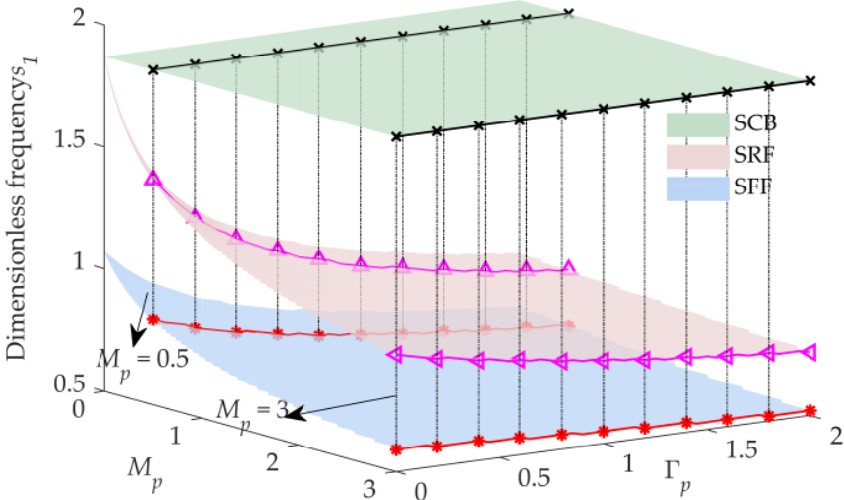

**Figure 6.** 3D surfaces of the first frequency versus $M_p$ and $\Gamma_p$ ($k_m = 0.5$).

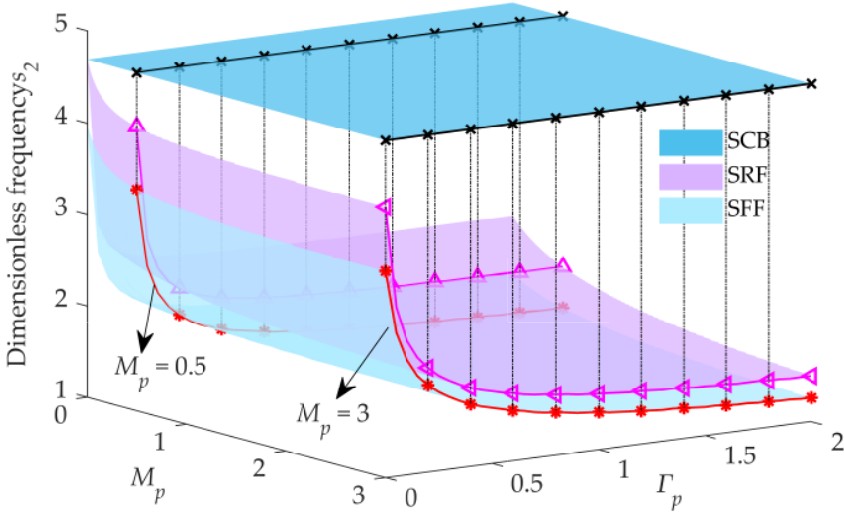

**Figure 7.** 3D surfaces of the second frequency versus $M_p$ and $\Gamma_p$ ($k_m = 0.5$).

Therefore, due to joint flexibility, the frequencies of SFF are lower than those of SCB and SRF. Moreover, if the stiffness of the joint is large and the moment of inertia in the tip load is small, the differences in the lower-order frequencies among SCB, SRF, and SFF are not significant. It can be inferred that in this case, whichever mode is selected as the assumed mode has no obvious difference in the model. However, in the opposite case, the differences in lower-order frequencies among SCB, SRF, and SFF are significant. Thus, it is necessary to be careful in choosing the assumed mode. Hence, the corresponding dynamic responses are studied in the next section.

### 2.3. Dynamic Response of a Flexible Space Manipulator System

The dynamic model of the space manipulator system has been established in Section 2.1. A simulation of the flexible two-link manipulator system was performed using MATLAB to analyze the dynamic response in this section. As mentioned in the previous section, to compare the difference in dynamic responses among the three assumed modes, the parameter $k_m$ should be small. The parameters of the system are listed in Table 2. Both flexible joint motors are commanded to output the following sinusoidal force,

$$\tau = \begin{cases} \sin(\pi t)N, 0\,\text{s} \le t \le 2\,\text{s} \\ 0N, 2\,\text{s} < t \le 10\,\text{s} \end{cases} \tag{35}$$

**Table 2.** The values of system parameters.

| Symbol | Value | Link1 | Link2 |
|--------|-------|-------|-------|
| $m_0$ | 17.23 kg | \ | \ |
| $m_p$ | 2.0 kg | \ | \ |
| $J_0$ | 0.087 kg $\cdot$ m$^2$ | \ | \ |
| $J_p$ | 0.005 kg $\cdot$ m$^2$ | \ | \ |
| $E_1, E_2$ | \ | 72.0 GPa | 72.0 GPa |
| $l_0$ | 0.12 m | \ | \ |
| $\rho_1, \rho_2$ | \ | 1.620 kg/m | 1.620 kg/m |
| $J_1, J_2$ | \ | 0.005 kg $\cdot$ m$^2$ | 0.005 kg $\cdot$ m$^2$ |
| $k_1, k_2$ | \ | 500 Nm/rad | 500 Nm/rad |
| $I_1, I_2$ | \ | $4.50 \times 10^{-8}$ m$^4$ | $4.50 \times 10^{-8}$ m$^4$ |
| $l_1, l_2$ | \ | 2.0 m | 2.0 m |

The terminal deformations of the two links are shown in Figure 8. The results demonstrate that the dynamic response amplitude of SFF is the largest, although there is no significant difference in the dynamic response among the three assumed modes. The

single-sided amplitude spectrum results show that the low-order natural frequencies are also close, as shown in Figure 9. Therefore, even if $k_m$ is small, the selection of the three kinds of assumed modes has no significant influence on the dynamic response. Therefore, when the stiffness of the joint is difficult to obtain and the model accuracy requirement is not high, it is acceptable to choose the modes of SRF or SFF. However, if the stiffness of the joint can be accurately determined, the mode of SFF is more suitable because the boundary condition is more similar to the actual system. In addition, because of the coupled dynamic characteristics, the internal resonance phenomenon occurs between joint 1 and joint 2 after 2 s. Hence, higher requirements on the performance of the controller are raised.

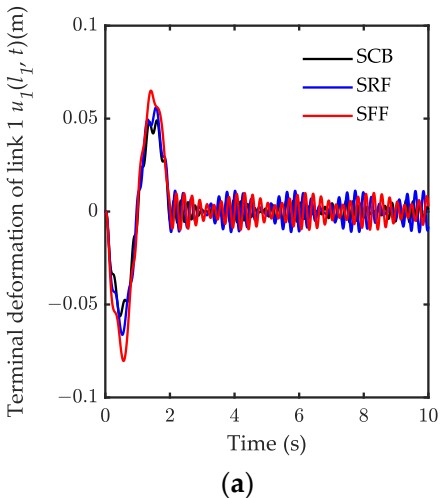
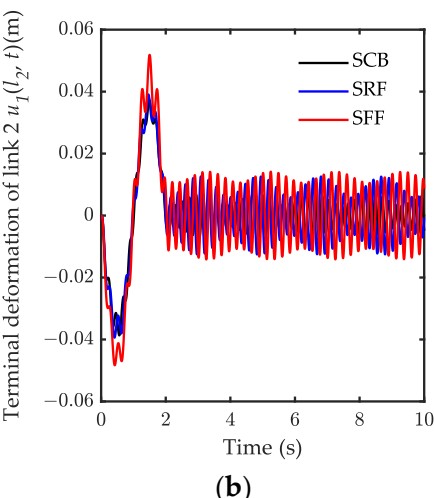

**Figure 8.** The terminal deformations of the two links: (**a**) the terminal deformation of link1; (**b**) the terminal deformation of link 2.

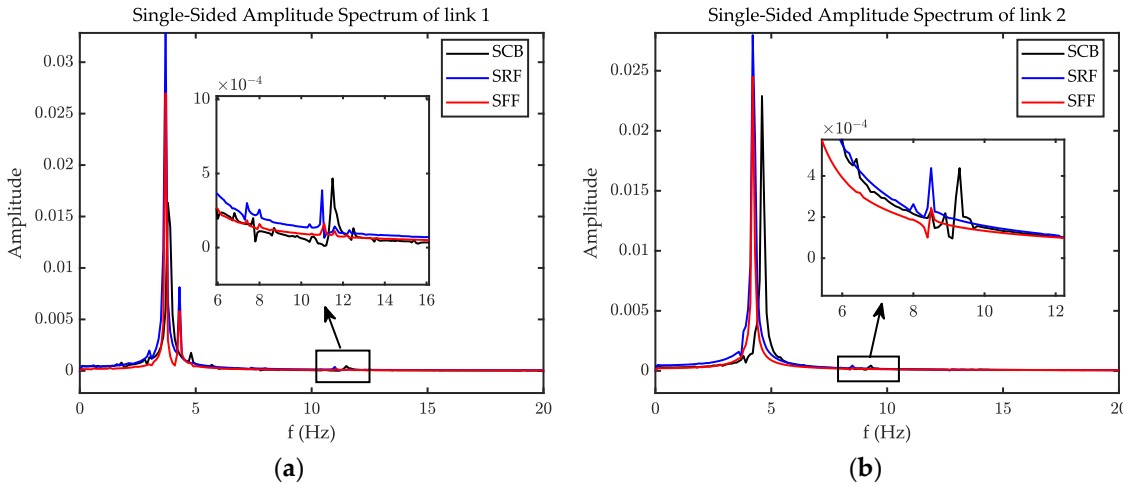

**Figure 9.** The single-sided amplitude spectrum: (**a**) the single-sided amplitude spectrum of link1; (**b**) the single-sided amplitude spectrum of link2.

## 3. Controller Design

### 3.1. Singular Perturbation Model of Flexible Space Manipulator System

When using the motor torque as the only input to control the output at the end of the flexible link, the system exhibits non-minimum phase characteristics [27]. Thus, it is not an easy task to suppress vibrations. In general, the rigid body motion of the system and the elastic vibration of the system occur in different timescales [28], and the frequency of rigid body motion is much less than one of the elastic vibration. Additionally, the system equation is nonlinear and coupled, thus, the calculation is hard to perform. Therefore,

based on the assumption that huge differences in the frequency domain, between the rigid motion and the elastic vibration, the singular perturbation method was introduced to decompose the system into a slow subsystem, which describes the rigid-body motion, and a fast subsystem, which describes the elastic vibration, after which a composite controller can be designed [28]. The dynamic model can be transformed into a singular perturbation model as described:

Define a matrix **D**

$$\mathbf{D} = \mathbf{M}^{-1} = \begin{bmatrix} \mathbf{D}_{11}(\boldsymbol{\theta}, \mathbf{q}) & \mathbf{D}_{12}(\boldsymbol{\theta}, \mathbf{q}) \\ \mathbf{D}_{21}(\boldsymbol{\theta}, \mathbf{q}) & \mathbf{D}_{22}(\boldsymbol{\theta}, \mathbf{q}) \end{bmatrix}. \tag{36}$$

From the system Equation (17), one can obtain:

$$\begin{bmatrix} \ddot{\mathbf{x}}_0 \\ \ddot{\boldsymbol{\theta}} \end{bmatrix} = -\mathbf{D}_{11}\mathbf{F}_1 - \mathbf{D}_{12}\mathbf{F}_2 - \mathbf{D}_{12}\mathbf{K}_q\mathbf{q} + \mathbf{D}_{11}\begin{bmatrix} \mathbf{0}_{3\times1} \\ \mathbf{K}_\sigma\boldsymbol{\sigma} \end{bmatrix} + \mathbf{D}_{12}\boldsymbol{\tau}', \tag{37}$$

$$\ddot{\mathbf{q}} = -\mathbf{D}_{21}\mathbf{F}_1 - \mathbf{D}_{22}\mathbf{F}_2 - \mathbf{D}_{22}\mathbf{K}_q\mathbf{q} + \mathbf{D}_{21}\begin{bmatrix} \mathbf{0}_{3\times1} \\ \mathbf{K}_\sigma\boldsymbol{\sigma} \end{bmatrix} + \mathbf{D}_{22}\boldsymbol{\tau}', \tag{38}$$

$$\ddot{\boldsymbol{\sigma}} = -\mathbf{J}^{-1}\mathbf{K}_\sigma\boldsymbol{\sigma} + \mathbf{J}^{-1}\boldsymbol{\tau} - \ddot{\boldsymbol{\theta}}_{sub}, \tag{39}$$

where $\sigma = \boldsymbol{\alpha} - \boldsymbol{\theta}_{sub}$. Define a singular perturbation factor $\varepsilon^2 = 1/\min(\mathbf{K}_\sigma, \mathbf{K}_q)$, and use the factor to define:

$$\overline{\mathbf{K}}_\sigma = \varepsilon^2\mathbf{K}_\sigma, \ \overline{\mathbf{K}}_q = \varepsilon^2\mathbf{K}_q, \ \mathbf{z}_\sigma = \boldsymbol{\sigma}\frac{1}{\varepsilon^2}, \ \mathbf{z}_q = \mathbf{q}\frac{1}{\varepsilon^2}. \tag{40}$$

From Equation (22), one obtains $\boldsymbol{\tau}' = \mathbf{K}_v\mathbf{U}$. Thus, $\mathbf{K}_v = \varepsilon\overline{\mathbf{K}}_v$ due to $O(K_a) = O(\varepsilon)$. Substituting Equation (40) into Equations (37)–(39) yields:

$$\varepsilon^2\ddot{\mathbf{z}}_\sigma = -\mathbf{J}^{-1}\overline{\mathbf{K}}_\sigma\mathbf{z}_\sigma + \mathbf{J}^{-1}\boldsymbol{\tau} - \ddot{\boldsymbol{\theta}}_{sub}, \tag{41}$$

$$\begin{bmatrix} \ddot{\mathbf{x}}_0 \\ \ddot{\boldsymbol{\theta}} \end{bmatrix} = -\mathbf{D}_{11}(\boldsymbol{\theta}, \varepsilon^2\mathbf{z}_q)\mathbf{F}_1(\boldsymbol{\theta}, \dot{\boldsymbol{\theta}}, \varepsilon^2\mathbf{z}_q, \varepsilon^2\dot{\mathbf{z}}_q) - \mathbf{D}_{12}(\boldsymbol{\theta}, \varepsilon^2\mathbf{z}_q)\mathbf{F}_2(\boldsymbol{\theta}, \dot{\boldsymbol{\theta}}, \varepsilon^2\mathbf{z}_q, \varepsilon^2\dot{\mathbf{z}}_q)$$
$$-\mathbf{D}_{12}(\boldsymbol{\theta}, \varepsilon^2\mathbf{z}_q)\overline{\mathbf{K}}_q\mathbf{z}_q + \mathbf{D}_{11}(\boldsymbol{\theta}, \varepsilon^2\mathbf{z}_q)\begin{bmatrix} \mathbf{0}_{3\times1} \\ \overline{\mathbf{K}}_\sigma\mathbf{z}_\sigma \end{bmatrix} + \varepsilon\mathbf{D}_{12}(\boldsymbol{\theta}, \varepsilon^2\mathbf{z}_q)\overline{\mathbf{K}}_v\mathbf{U} \tag{42}$$

$$\varepsilon^2\ddot{\mathbf{z}}_q = -\mathbf{D}_{21}(\boldsymbol{\theta}, \varepsilon^2\mathbf{z}_q)\mathbf{F}(\boldsymbol{\theta}, \dot{\boldsymbol{\theta}}, \varepsilon^2\mathbf{z}_q, \varepsilon^2\dot{\mathbf{z}}_q)_1 - \mathbf{D}_{22}(\boldsymbol{\theta}, \varepsilon^2\mathbf{z}_q)\mathbf{F}_2(\boldsymbol{\theta}, \dot{\boldsymbol{\theta}}, \varepsilon^2\mathbf{z}_q, \varepsilon^2\dot{\mathbf{z}}_q)$$
$$-\mathbf{D}_{22}(\boldsymbol{\theta}, \varepsilon^2\mathbf{z}_q)\overline{\mathbf{K}}_q\mathbf{z}_q + \mathbf{D}_{21}(\boldsymbol{\theta}, \varepsilon^2\mathbf{z}_q)\begin{bmatrix} \mathbf{0}_{3\times1} \\ \overline{\mathbf{K}}_\sigma\mathbf{z}_\sigma \end{bmatrix} + \varepsilon\mathbf{D}_{22}(\boldsymbol{\theta}, \varepsilon^2\mathbf{z}_q)\overline{\mathbf{K}}_v\mathbf{U} \tag{43}$$

where superscript "$\overline{\phantom{x}}$" indicates the value of the variables at $\varepsilon = 0$.

If $\varepsilon = 0$, from Equation (41), one can obtain:

$$\overline{\mathbf{z}}_\sigma = \overline{\mathbf{K}}_\sigma^{-1}(\boldsymbol{\tau}_s - \mathbf{J}\ddot{\overline{\boldsymbol{\theta}}}_{sub}). \tag{44}$$

In Equation (44), $\boldsymbol{\tau}_s$ is the value of $\boldsymbol{\tau}$ at $\varepsilon = 0$, which can also be written as $\overline{\boldsymbol{\tau}}$. Substituting Equation (44) into Equation (43) provides:

$$\overline{\mathbf{z}}_q = \overline{\mathbf{K}}_q^{-1}\overline{\mathbf{D}}_{22}^{-1}(\overline{\boldsymbol{\theta}}, \mathbf{0})\left[-\overline{\mathbf{D}}_{21}(\overline{\boldsymbol{\theta}}, \mathbf{0})\overline{\mathbf{F}}_1(\overline{\boldsymbol{\theta}}, \dot{\overline{\boldsymbol{\theta}}}, \mathbf{0}, \mathbf{0}) - \overline{\mathbf{D}}_{22}(\overline{\boldsymbol{\theta}}, \mathbf{0})\overline{\mathbf{F}}_2(\overline{\boldsymbol{\theta}}, \dot{\overline{\boldsymbol{\theta}}}, \mathbf{0}, \mathbf{0}) + \overline{\mathbf{D}}_{21}(\overline{\boldsymbol{\theta}}, \mathbf{0})\begin{bmatrix} \mathbf{0}_{3\times1} \\ \overline{\mathbf{K}}_\sigma\mathbf{z}_\sigma \end{bmatrix}\right). \tag{45}$$

Finally, substituting Equations (44) and (45) into Equation (42) and using the inverse formula yields:

$$
\begin{bmatrix} \ddot{\overline{\mathbf{x}}}_0 \\ \ddot{\overline{\boldsymbol{\theta}}} \end{bmatrix} = (\overline{\mathbf{D}}_{11}(\overline{\boldsymbol{\theta}}) - \overline{\mathbf{D}}_{12}(\overline{\boldsymbol{\theta}})\overline{\mathbf{D}}_{22}^{-1}(\overline{\boldsymbol{\theta}})\overline{\mathbf{D}}_{21}(\overline{\boldsymbol{\theta}}))\left( \begin{bmatrix} \mathbf{0}_{3\times 1} \\ \boldsymbol{\tau}_s \end{bmatrix} - \begin{bmatrix} \mathbf{0}_{3\times 1} \\ \mathbf{J}\ddot{\overline{\boldsymbol{\theta}}}_{sub} \end{bmatrix} - \overline{\mathbf{F}}_1(\overline{\boldsymbol{\theta}}, \dot{\overline{\boldsymbol{\theta}}}) \right). \tag{46}
$$

Equation (46) is the quasi-steady-state equation for the system and the slow subsystem. Defining the following boundary layer correction terms as:

$$
\boldsymbol{\eta}_1 = \mathbf{z}_\sigma - \overline{\mathbf{z}}_\sigma, \boldsymbol{\eta}_2 = \varepsilon \dot{\mathbf{z}}_\sigma, \boldsymbol{\beta}_1 = \mathbf{z}_q - \overline{\mathbf{z}}_q, \boldsymbol{\beta}_2 = \varepsilon \dot{\mathbf{z}}_q. \tag{47}
$$

Substituting Equation (47) into Equations (41)–(43) gives the fast subsystem equation,

$$
\frac{d\boldsymbol{\eta}}{d\gamma} = \mathbf{A}_f \boldsymbol{\eta} + \mathbf{B}_f \mathbf{U}_f, \tag{48}
$$

where $\boldsymbol{\eta} = \begin{bmatrix} \boldsymbol{\eta}_1 \\ \boldsymbol{\beta}_1 \\ \boldsymbol{\eta}_2 \\ \boldsymbol{\beta}_2 \end{bmatrix}, \mathbf{U}_f = \begin{bmatrix} \boldsymbol{\tau}_f \\ \boldsymbol{\tau}' \end{bmatrix}, \mathbf{A}_f = \begin{bmatrix} \mathbf{0} & \mathbf{0} & \mathbf{I} \\ -\mathbf{J}^{-1}\overline{\mathbf{K}}_\sigma & \mathbf{0} & \mathbf{0} \\ \overline{\mathbf{D}}_{21_{12}}\overline{\mathbf{K}}_\sigma & -\overline{\mathbf{D}}_{22}\overline{\mathbf{K}}_q & \mathbf{0} \end{bmatrix}, \mathbf{B}_f = \begin{bmatrix} \mathbf{0} & \mathbf{0} \\ \mathbf{J}^{-1} & \mathbf{0} \\ \mathbf{0} & \overline{\mathbf{D}}_{22} \end{bmatrix}, \overline{\mathbf{D}}_{21_{12}}$

is a submatrix of $\overline{\mathbf{D}}_{22}$, which does not contain elements in the first column of the matrix $\overline{\mathbf{D}}_{22}$.

### 3.2. Computed Torque Controller Designed for Joints

The system of space manipulators with a freebase is underactuated. To design the underactuated computed torque controller, the parameter $\delta$ is introduced. Based on the slow subsystem Equation (46), the control torque was designed as:

$$
\begin{bmatrix} \mathbf{0} \\ \boldsymbol{\tau}_s \end{bmatrix} = \left( \overline{\mathbf{M}}_{11}(\overline{\boldsymbol{\theta}}) + \begin{bmatrix} \mathbf{0} & \\ & \mathbf{J} \end{bmatrix} \right) \begin{bmatrix} \boldsymbol{\delta} \\ \mathbf{u}_s \end{bmatrix} + \overline{\mathbf{F}}_1(\overline{\boldsymbol{\theta}}, \dot{\overline{\boldsymbol{\theta}}}), \tag{49}
$$

where $\mathbf{u}_s$ is the reference input, determined by the outer loop control. Substituting Equation (49) into Equation (46) provides:

$$
\begin{bmatrix} \boldsymbol{\delta} \\ \mathbf{u}_s \end{bmatrix} = \begin{bmatrix} \ddot{\overline{\mathbf{x}}}_0 \\ \ddot{\overline{\boldsymbol{\theta}}} \end{bmatrix} = \begin{bmatrix} \ddot{\overline{\mathbf{x}}}_0 \\ \ddot{\overline{\theta}}_0 \\ \ddot{\overline{\theta}}_{sub} \end{bmatrix}. \tag{50}
$$

In this way, the inner loop control is completed by introducing model-based torque. A PD controller was introduced, and the reference input $\mathbf{u}_s$ was obtained by:

$$
\mathbf{u}_s = -\mathbf{K}_p \mathbf{e}_s - \mathbf{K}_d \dot{\mathbf{e}}_s + \ddot{\boldsymbol{\theta}}_d, \tag{51}
$$

where $\mathbf{K}_p$ and $\mathbf{K}_d$ are the position feedback gain matrix and velocity feedback gain matrix, respectively, both of which are positive definite, and $\mathbf{e}_s = \boldsymbol{\theta}_d - \overline{\boldsymbol{\theta}}_{sub}$ is the error between desired position $\boldsymbol{\theta}_d$, and actual output joint position $\overline{\boldsymbol{\theta}}_{sub}$. From Equation (44), one can easily deduce that the variable $\overline{\mathbf{z}}_\sigma$ is bounded, which indicates the elastic deformation vector $\overline{\boldsymbol{\sigma}} = 0$ at $\varepsilon = 0$ (Since $\mathbf{z}_\sigma = \boldsymbol{\sigma}\frac{1}{\varepsilon^2}$, and $\overline{\mathbf{z}}_\sigma$ is bounded at $\varepsilon = 0$, then, $\overline{\boldsymbol{\sigma}} = 0$ at $\varepsilon = 0$ necessarily). Therefore, $\overline{\boldsymbol{\theta}}_{sub} = \overline{\boldsymbol{\alpha}}$, $\mathbf{e}_s = \boldsymbol{\theta}_d - \overline{\boldsymbol{\alpha}}$, motor position can be used for the inner loop control.

Substituting Equation (51) into Equation (50) yields:

$$
\begin{cases} \boldsymbol{\delta} = \begin{bmatrix} \ddot{\overline{\mathbf{x}}}_0 \\ \ddot{\overline{\theta}}_0 \end{bmatrix} \\ \ddot{\mathbf{e}}_s + \mathbf{K}_p \mathbf{e}_s + \mathbf{K}_v \dot{\mathbf{e}}_s = \mathbf{0} \end{cases}, \tag{52}
$$

where $(\mathbf{e}_s, \dot{\mathbf{e}}_s) = (\mathbf{0}, \mathbf{0})$ is the globally asymptotically stable equilibrium point since $\mathbf{K}_p$, $\mathbf{K}_d$ are positive definite, and $\delta$ represents the perturbed acceleration of the base body, whereby the effect cannot be ignored for a small mass base.

$\delta$ can be solved from Equation (49):

$$\delta = \overline{\mathbf{M}}^{-1}{}_{11_{11}}(-\overline{\mathbf{M}}_{11_{12}}\mathbf{u}_s - \overline{\mathbf{F}}_1(\theta, \dot{\theta})), \tag{53}$$

where $\overline{\mathbf{M}}_{11_{11}} = \overline{\mathbf{M}}(1:3, 1:3)$, $\overline{\mathbf{M}}_{11_{12}} = \overline{\mathbf{M}}(1:3, 4:n+1)$.

### 3.3. Adaptive Fuzzy Controller Designed for Piezo Actuator

The system decomposed into a slow subsystem and a fast subsystem, as noted in Section 3.1. The fast subsystem, Equation (48), is linear. The LQR controller introduced for the linear systems by researchers [28] may be used to suppress the vibrations. However, the effectiveness of the LQR controller designed for fast subsystems relies heavily on modeling accuracy. In practice, joint friction, structural damping, etc., are difficult to be modeled. Furthermore, the modal truncation introduced by the assumed mode method reduces the accuracy of the model. Therefore, a direct adaptive fuzzy controller is presented to suppress the vibration of the flexible links.

The fuzzy system of a space manipulator system can be described as $\hat{\mathbf{F}}(\mathbf{q}|\boldsymbol{\gamma})$. A fuzzy controller is designed by using product inference engine, gauss fuzzier, and a center averaging defuzzifier. According to the controller design method, based on a traditional fuzzy system, the robust fuzzy adaptive control law is designed as:

$$\boldsymbol{\tau}'_{mn \times 1} = \hat{\mathbf{F}}(\mathbf{q}|\boldsymbol{\gamma}) - \mathbf{K}_D\mathbf{s} - \mathbf{W}\text{sgn}(\mathbf{s}), \tag{54}$$

where $\mathbf{s} = d\mathbf{e}_f + \Lambda\mathbf{e}_f$, and $\mathbf{K}_D$, $\mathbf{W}$, and $\Lambda$ are weight matrices. The sign function in Equation (54) is designed to address the problem of external disturbances. However, the sign function may lead to a high-frequency chattering phenomenon. To avoid high-frequency chattering, the sign function can be substituted for by the saturation function. Thus, the adaptive fuzzy controller is designed as:

$$\boldsymbol{\tau}'_{mn \times 1} = \hat{\mathbf{F}}(\mathbf{q}, \dot{\mathbf{q}}|\boldsymbol{\gamma}) - \mathbf{K}_D\mathbf{s} - \mathbf{W}sat(\mathbf{s}), \tag{55}$$

where $sat(\mathbf{s}) = \begin{cases} 1, \mathbf{s} > \Delta \\ \frac{1}{\Delta}\mathbf{s}, |\mathbf{s}| \leq \Delta \\ -1, \mathbf{s} < \Delta \end{cases}$, and the parameter $\Delta$ is generally set to a small value.

The adaptive law is designed as:

$$\dot{\boldsymbol{\gamma}}_i = -\zeta_i^{-1}\mathbf{s}_i\xi(\dot{\mathbf{q}}), \; i = 1, 2, \cdots, n, \tag{56}$$

where $\zeta_i$ ($\zeta_i > 0$) is called the adaptive parameter.

The fuzzy system is designed as:

$$\hat{\mathbf{F}}(\dot{\mathbf{q}}|\boldsymbol{\gamma}) = \begin{bmatrix} \hat{F}_1(\dot{q}_1^1) \\ \vdots \\ \hat{F}_2(\dot{q}_1^m) \\ \vdots \\ \hat{F}_{m \times n}(\dot{q}_n^m) \end{bmatrix} = \begin{bmatrix} \boldsymbol{\gamma}_1^T\xi^1(\dot{q}_1^1) \\ \vdots \\ \boldsymbol{\gamma}_2^T\xi^2(\dot{q}_1^m) \\ \vdots \\ \boldsymbol{\gamma}_{m \times n}^T\xi^{m \times n}(\dot{q}_n^m) \end{bmatrix}. \tag{57}$$

The basic structure of the presented fuzzy controller is shown in Figure 10. The link end displacement $\mathbf{e}_f$ and its speed $d\mathbf{e}_f$ are used as control inputs, while $\mathbf{E}_f$ and $d\mathbf{E}_f$ are the fuzzy quantities corresponding to the two inputs. The control torque $\tau_f$ of the piezoelectric actuator is obtained by defuzzing the fuzzy quantity $M$, obtained by the fuzzy logic inference.

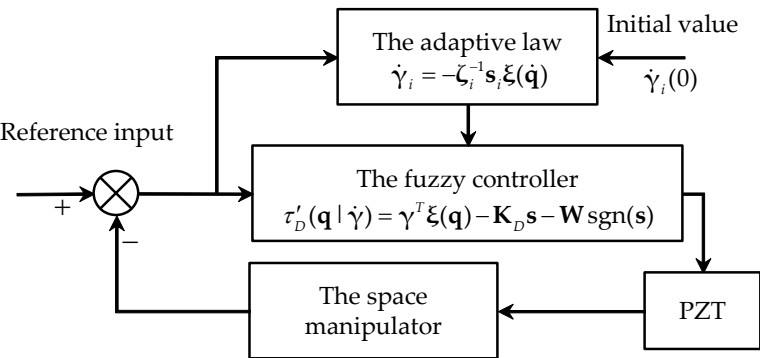

**Figure 10.** Schematic of adaptive fuzzy controller.

## 4. Numerical Simulation

The effectiveness of the presented control strategy is examined on a two-link flexible manipulator system, as shown in Figure 11. A pair of PZT actuators are attached to the root of each flexible link and a tip payload is attached to the distal end of the second flexible link. The main parameters of the manipulator system are listed in Table 3.

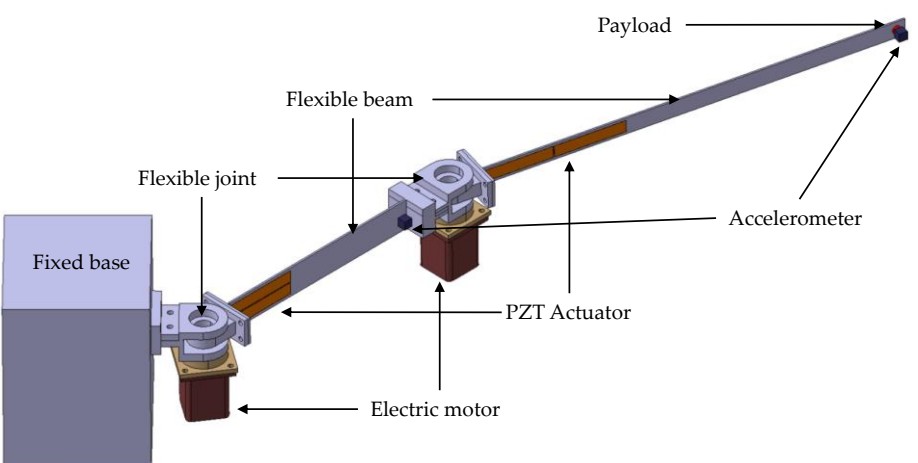

**Figure 11.** Two-link flexible manipulator with fixed base.

**Table 3.** Kinematic and dynamic parameters of the space manipulator system.

| Symbol | Value | Link1 | Link2 |
|---|---|---|---|
| $m_p$ | 0.1 kg | \ | \ |
| $J_p$ | $1.25 \times 10^{-6}$ kg $\cdot$ m$^2$ | \ | \ |
| $E_1, E_2$ | \ | 200.0 GPa | 200.0 GPa |
| $\rho_1, \rho_2$ | \ | 0.5688 kg/m | 0.3160 kg/m |
| $J_1, J_2$ | \ | 0.001 kg $\cdot$ m$^2$ | 0.001 kg $\cdot$ m$^2$ |
| $k_1, k_2$ | \ | 9.6 Nm/rad | 9.6 Nm/rad |
| $I_1, I_2$ | \ | $2.40 \times 10^{-11}$ m$^4$ | $1.33 \times 10^{-11}$ m$^4$ |
| $l_1, l_2$ | \ | 0.5 m | 0.25 m |

The membership functions of $\mathbf{e}_f$ and $\mathrm{d}\mathbf{e}_f$ used in the two cases are shown in Figures 12 and 13, respectively. Define nine levels of fuzzy value: PB (positive big), PM (positive middle), PS (positive small), PO (positive zero), ZO (zero), NO (negative zero), NS (negative small), NM (negative middle), and NB (negative big).

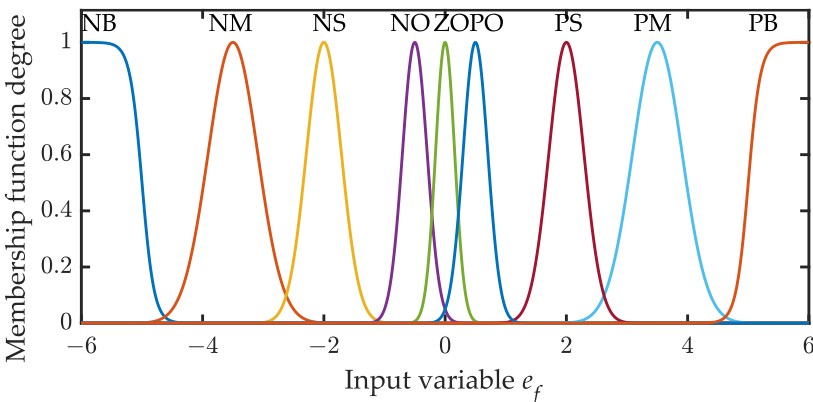

**Figure 12.** Membership function of input variable $\mathbf{e}_f$.

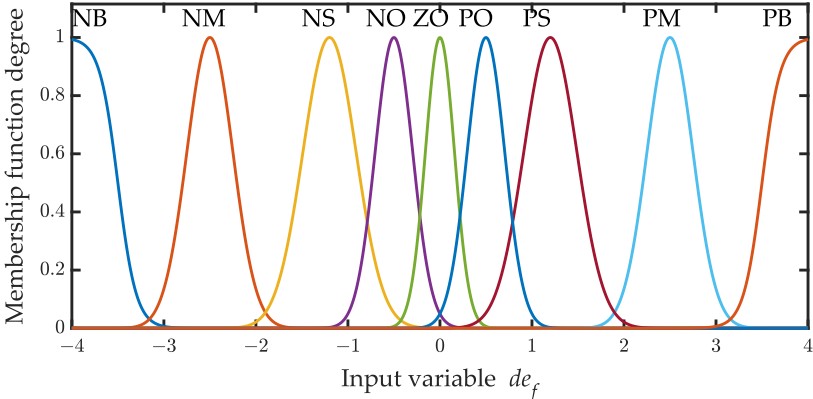

**Figure 13.** Membership function of input variable d$\mathbf{e}_f$.

In this section, the performance of the proposed control scheme is evaluated for two cases: setpoint motion control and periodic motion control. The PD and CTM controllers were firstly used to track the joint trajectories with their performance comparisons. The PD controller and AFC were, then, adopted to suppress the elastic vibration of the links and were compared with each other. Moreover, to ensure the fairness of the comparisons, the parameters $\mathbf{K}_p$ and $\mathbf{K}_d$ in the inner loop controller of CTM were set to the same as the proportional control gain and differential control gain of the PD controller, respectively, and the performance of AFC and one PD were compared when the maximum voltage of PZT was limited. A set of gains for the PD controller for better control performance was tuned by numerical simulation.

*4.1. Case 1: Setpoint Motion Control*

The manipulator is commanded to track a desired cycloid trajectory to a desired position. The desired position of both links was set to $\pi/2$ rad, and the required time for the two links to reach the desired position was set to 5.0 s. The desired cycloid trajectory is $\theta_d = \pi \times [t/5 - \sin(2\pi t/5)/2\pi]/2$. The tip vibration caused by the motor stopping is called residual vibration. In order to demonstrate the control effect on the residual vibration, the simulation time was set to 10 s.

The comparison of tracking the control performance between the proposed computed torque method and the PD controller is shown in Figure 14. Figure 14b,e show the superior performance of CTM in tracking the desired trajectory, especially at the inflection point of the trajectory. Moreover, the output torque of CTM was smoother than that of PD, as evidenced in Figure 14c,f. The simulation result of acceleration response is shown in Figure 15. The amplitude of the residual vibration with AFC costs only 0.1 s to attenuate to 10% of its maximum amplitude, the PD controller costs 0.2 s, and uncontrol costs 0.18. The

AFC provides better performance than PD in rapidly suppressing the residual vibration. Additionally, compared to the uncontrol, the RMS of the acceleration response with AFC attenuated by 95%, whereas one with the PD controller only attenuated by 23%. The residual vibration attenuates under the uncontrol condition was due to the joint control torques, which kept the joint positions stable.

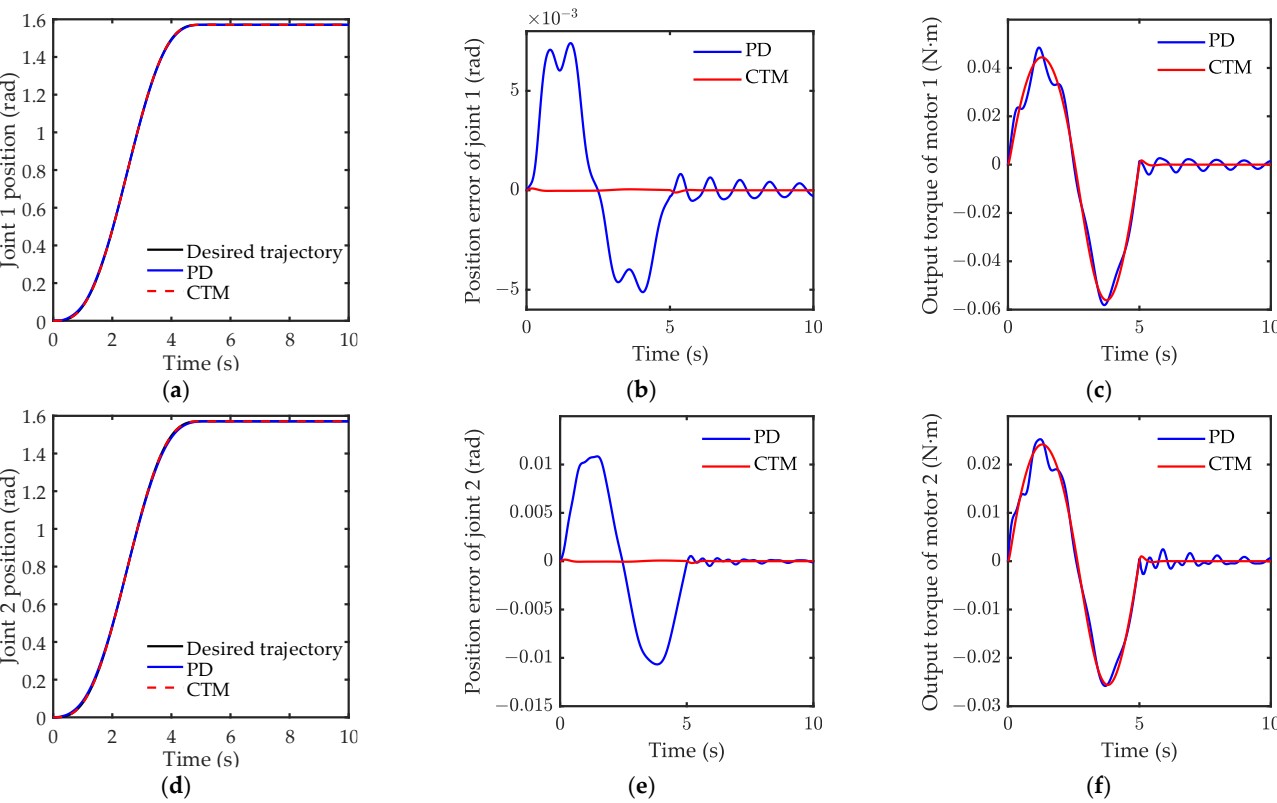

**Figure 14.** Simulation result of motion case 1: (**a**) joint 1 position; (**b**) position error of joint 1; (**c**) output torque of motor 1; (**d**) joint 2 position; (**e**) position error of joint 2; (**f**) output torque of motor 2.

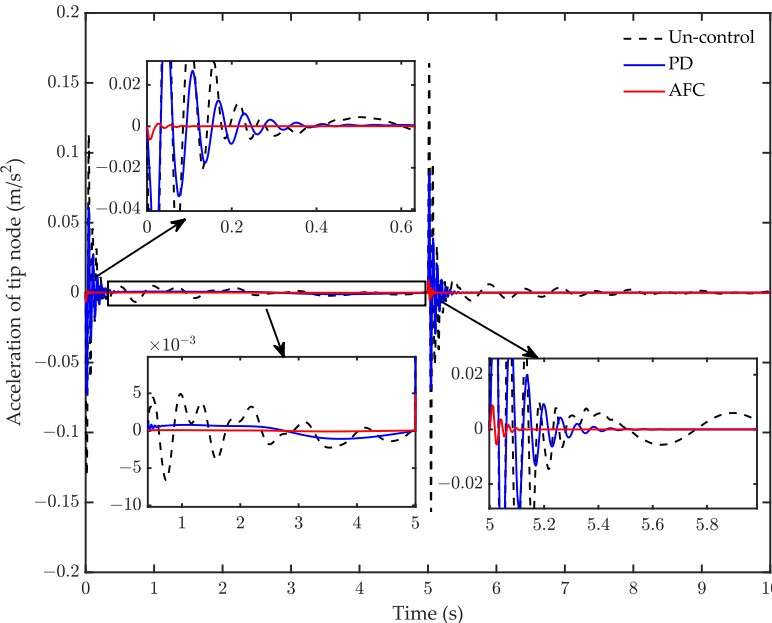

**Figure 15.** Simulation result of acceleration response for case 1.

### 4.2. Case 2: Periodic Motion Control

In case 2, a desired period sinusoidal trajectory $\theta_d(t) = A_m \sin(2\pi f_1 t + b) + A_m/2$ is selected to test the effectiveness of the proposed controller in a periodic motion, where $A_m = \pi/4$, $f_1 = 0.1$, and $b = 0$. The motion time was 30 s and the simulation time was 35 s. A comparison of tracking is shown in Figure 16. Here, CTM is seen to still provide better performance than PD in tracking the periodic motion. The trajectory of PD obviously buffets near the desired trajectory. It is worth noting that the tracking error of CTM visibly increases at 30 s, yet rapidly reduces after 30 s. Figure 16c,f show the torques of the two joints, respectively. It can be seen that the torques of the CTM are obviously smaller than for PD. Figure 17 shows a comparison of the control effect on the tip vibrational acceleration. Compared to the uncontrol, the RMS of the acceleration response with AFC attenuated by 95%, and one with the PD controller only attenuated by 62%. Obviously, AFC maintains a good performance in suppressing the vibration. In addition, in the two cases, the frequency of the elastic vibration was greater than 70 Hz, and one for the rigid motion was less than 3 Hz, which indicates that the elastic vibration and the rigid body motion occur in different timescales, hence, the introduction of the singular perturbation theory is suitable. The elastic deformation is very small in numerical simulation, which demonstrates that the small deformation assumption of the flexible joint is suitable.

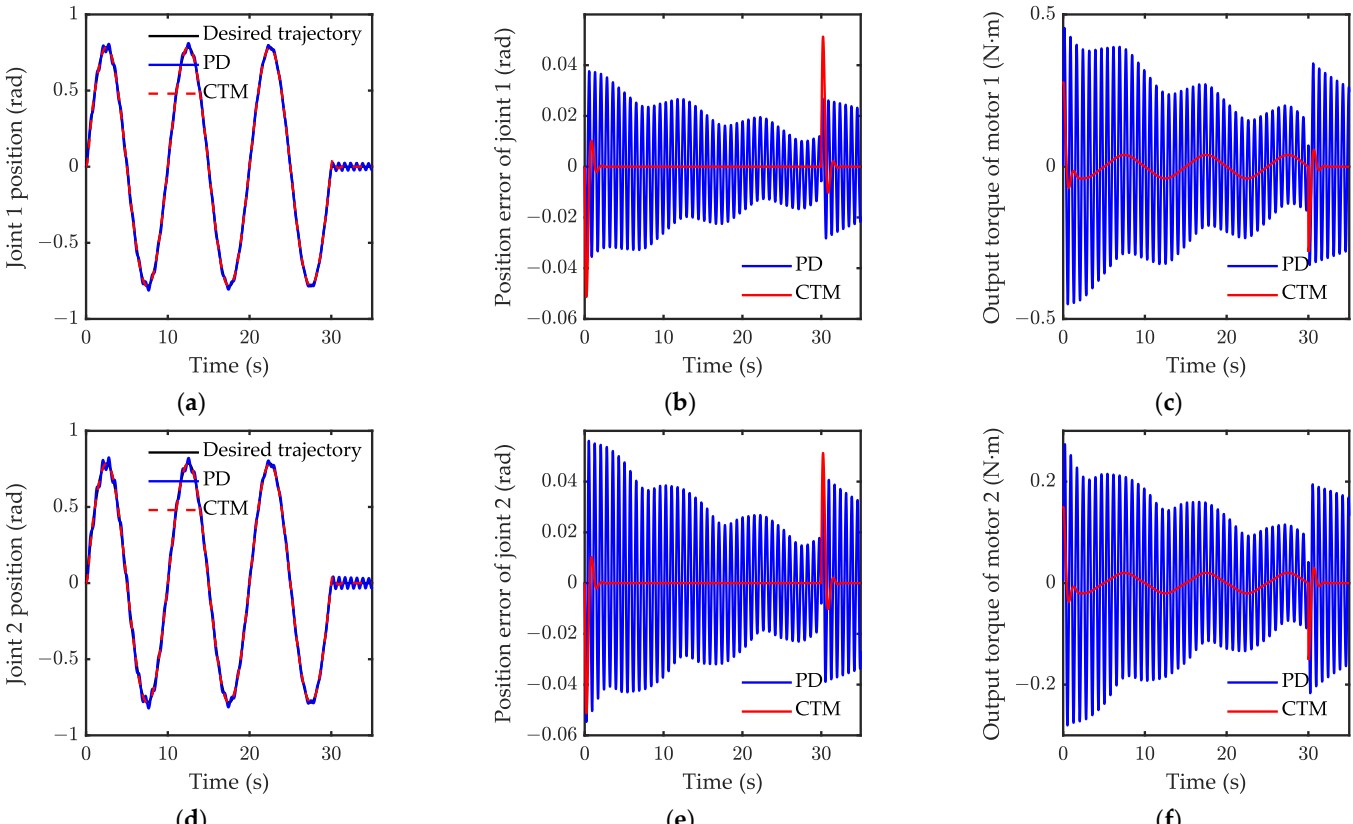

**Figure 16.** Simulation result of motion for case 2: (**a**) joint 1 position; (**b**) position error of joint 1; (**c**) output torque of motor 1; (**d**) joint 2 position; (**e**) position error of joint 2; (**f**) output torque of motor 2.

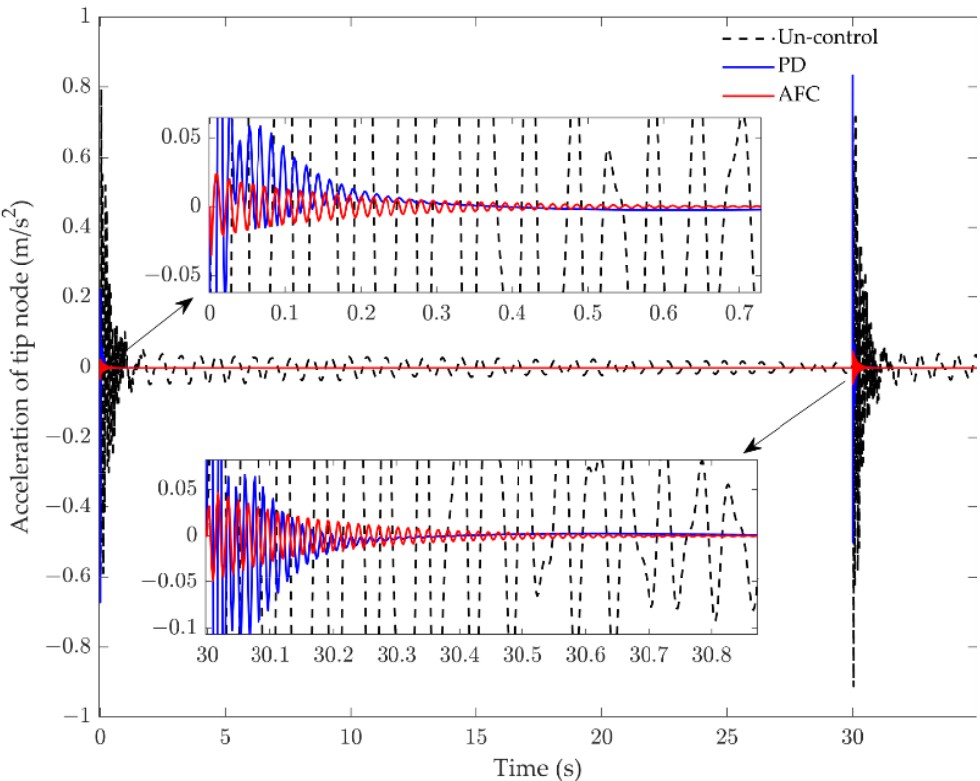

**Figure 17.** Simulation result of acceleration response for case 2.

## 5. Experimental Results

Experiments were performed to verify the simulations in case 1 and case 2. The experimental setup is schematically shown in Figure 18. The flexible links are clamped at the shafts of the motors through couplings, and the diagonal rope is supported on each joint to counter the gravity on the flexible link. Two AC rotary servomotors (designed by YASKAWA and SGM7J-01AFC6S) were used to drive the flexible links, and a built-in 24-bit absolute encoder installed within the motor was applied to calculate the rotating angle. Each flexible link had a PZT actuator (MFC, M-8514-P1) attached to the root, and the PZT actuators were used to suppress the elastic vibration on the flexible links. Each flexible link had an accelerometer (PCB 333B32) attached at the end to collect the vibration acceleration signal. The velocity signal and displacement signal were obtained using the first and second integration of the acceleration signal, respectively. In practice, the zero drift and high-frequency noise of the sensor cannot be avoided. Hence, a low-pass digital filter was designed to address the issue. A small-time delay could be introduced by the digital filter. However, the time delay was accepted in the experiment. Furthermore, the ambient laboratory temperature was maintained at a constant value to reduce the drift caused by the temperature changes. The main parameters of the experimental model are shown in Table 3. A motion controller (GALIL DMC1846) interfaced with a high-performance PC was used to snatch data and process data. The control voltage signals for the motor and the PZT actuators were sent to the servo driver and HVPZT amplifier using the GALIL motion controller, respectively. To remove the effect of the residual modes and high-frequency noise, a low-pass digital filter was designed for the vibrational acceleration signal.

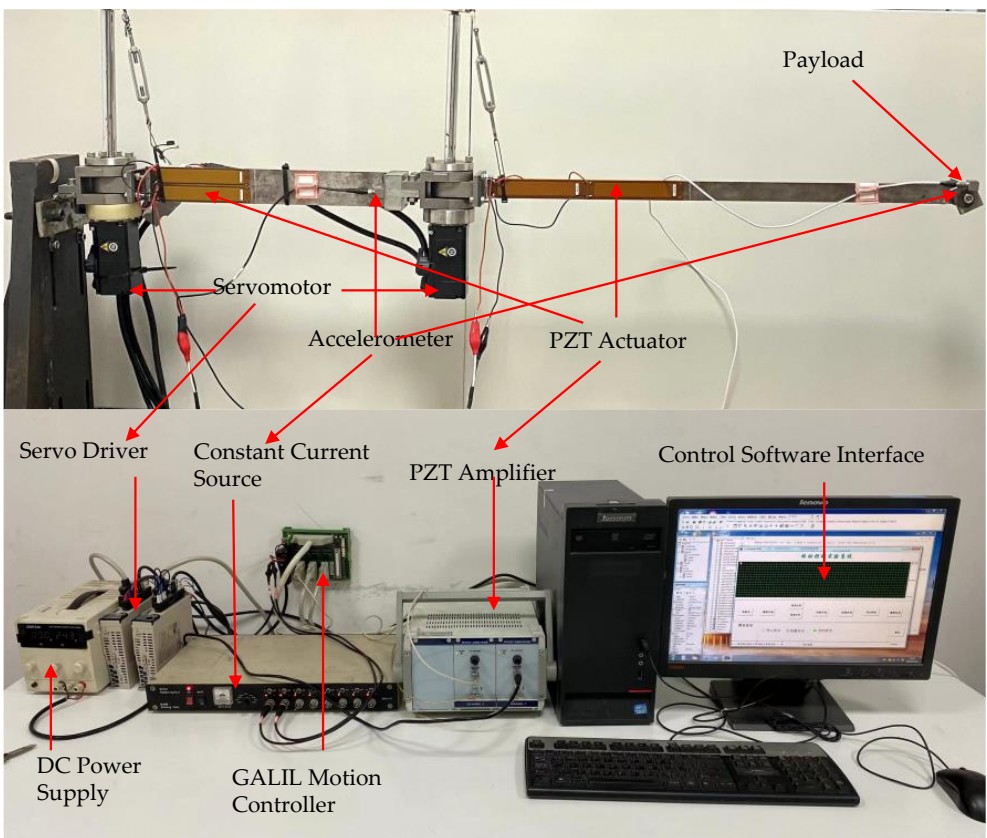

**Figure 18.** Scheme of the two-link flexible manipulator experimental setup.

The experimental results of the setpoint motion control are shown in Figures 19 and 20. Figure 19 shows that the CTM performs better than PD in the experiment. Figure 20 indicates that both PD and AFC can suppress residual vibration, although AFC had a better performance than the PD controller in suppressing the residual vibrations. The root mean square (RMS) and the amplitude of the whole course corresponding responses are listed in Table 4. Compared to the uncontrol, the RMS of the acceleration response with AFC was attenuated by 78.3%, while one with the PD controller attenuated by only 44.9%. This indicates that AFC provides better performance in suppressing the vibrations. Figure 19a,b demonstrate that serious buffet occurs under PD control, and the position error of PD is larger than for CTM. The position error of joint 2 is larger than for joint 1 under PD control during the motion, as shown in Figure 19c,d. The experimental performance of CTM was worse than the simulated performance. The position error was more significant during the first 3 s. CTM cannot perform high-precision trajectory tracking in the experiment. A possible reason is that the mathematical model does not consider the influence of nonlinear factors, such as friction and clearance, in the actual system.

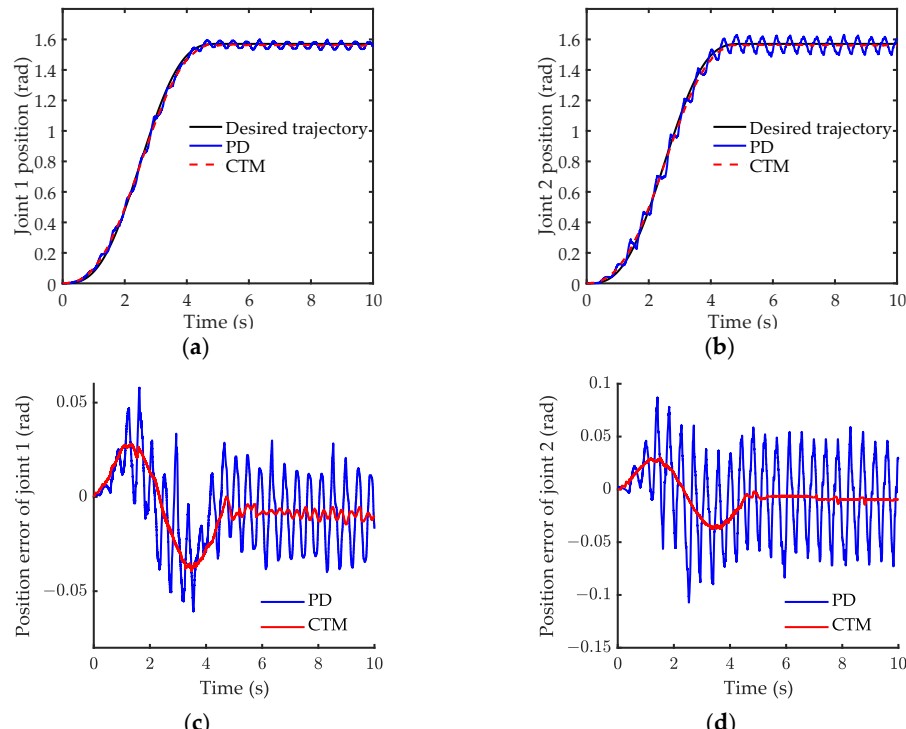

**Figure 19.** Experiment result of joint 1 position for case 1: (**a**) joint 1 position; (**b**) joint 2 position; (**c**) position error of joint 1; (**d**) position error of joint 2.

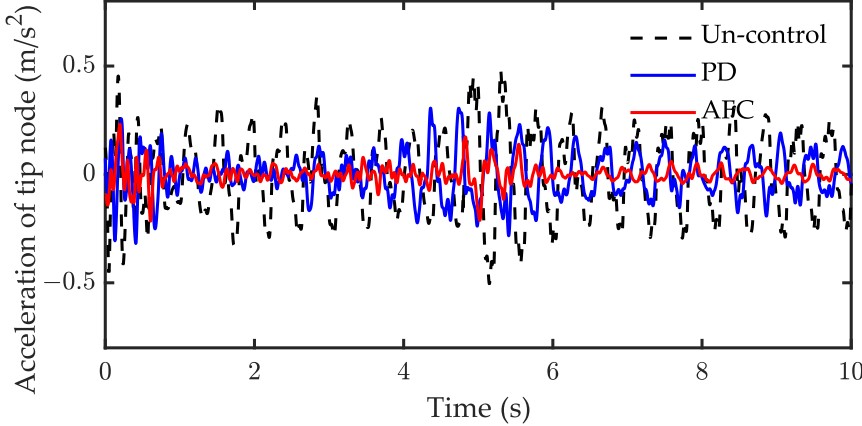

**Figure 20.** Experiment result of acceleration response for case 1.

**Table 4.** Vibration suppression effect of the experimental system for case 1.

| Controller | Uncontrol | PD | AFC |
|:---:|:---:|:---:|:---:|
| RMS (m/s$^2$) | 0.1847 | 0.1017 | 0.04 |
| Amplitude (m/s$^2$) | 0.5048 | 0.3186 | 0.2309 |

Experimental results under periodic motion control are shown in Figures 21–24, respectively. The results of the tracking shown in Figure 21 demonstrate that the joint positions are tracked well once the CTM is employed. Figure 22 shows the acceleration response of the tip node when the adaptive parameter is $\zeta_1 = \zeta_2 = \zeta = 2000$. The RMS and the amplitude of the whole course response are listed in Table 5. Both AFC and PD can suppress the vibration of the tip node. Compared with uncontrol, the RMS of AFC was attenuated by 62.5% and one for the PD controller was attenuated by 48.9%. In this case, the control performance of AFC was slightly better than of PD. The value of the adaptive parameter $\zeta$

is important for the control performance of AFC. The experimental results for the further adjustment of the parameters $\zeta$ are shown in Figures 23 and 24. Figures 23 and 24 show the control performance of AFC when $\zeta = 5000$ and $\zeta = 20,000$, respectively. When $\zeta = 5000$, the vibration is rapidly suppressed by employing AFC, the RMS of AFC was attenuated by 81.3% compared with the uncontrol. However, increasing the adaptive parameter to $\zeta = 20,000$ does not necessarily improve the performance of the AFC, the RMS of AFC was only attenuated by 57.1% compared to the uncontrol. When the value of the adaptive parameter $\zeta$ was small, the convergence of the AFC algorithm was slow and, thus, the AFC performance was poor. Overall, the acceleration can be suppressed quicker as the adaptive parameter increases. However, when the value of the adaptive parameter $\zeta$ was too large, the gain of the AFC was large, resulting in instability and causing additional vibrations. In light of these results on simulation, $4000 < \zeta < 6000$ is suitable for this experimental system.

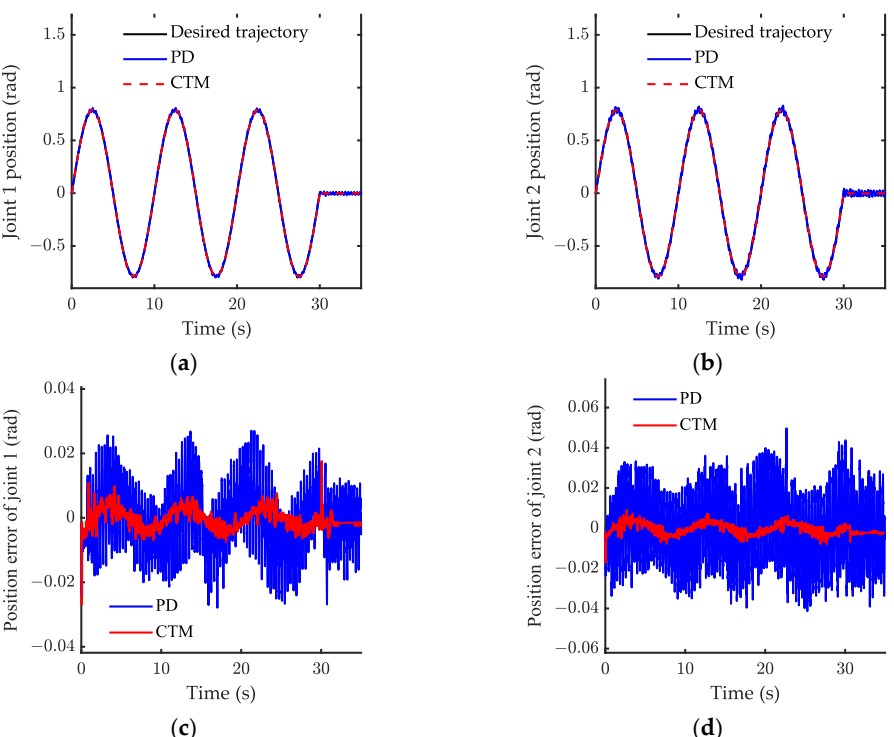

**Figure 21.** Experiment result of joint 1 position for case 2: (**a**) joint 1 position; (**b**) joint 2 position; (**c**) position error of joint 1; (**d**) position error of joint 2.

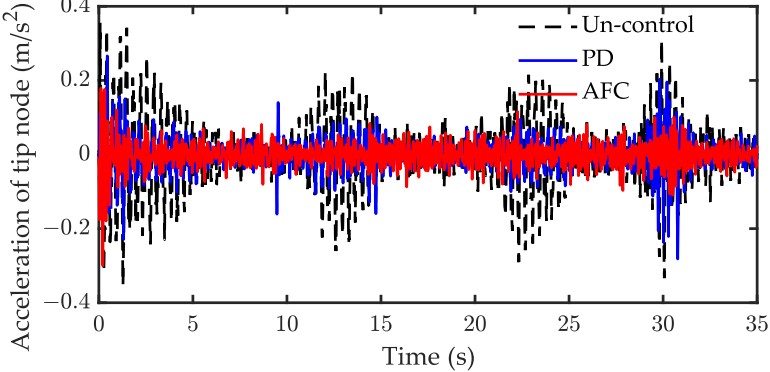

**Figure 22.** Experiment result of acceleration response for case 2 ($\zeta = 2000$).

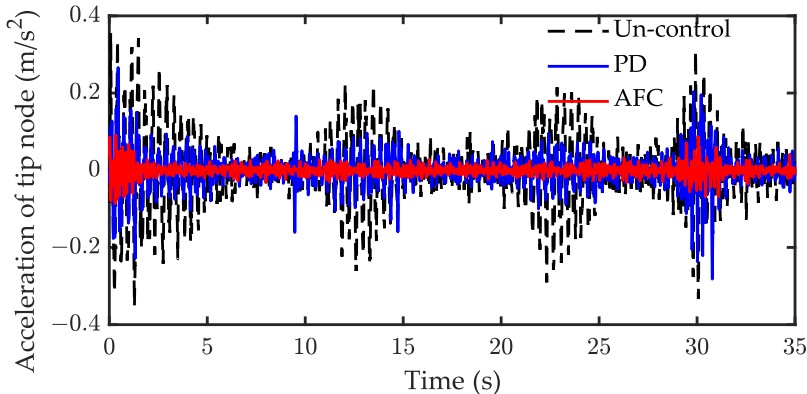

**Figure 23.** Experiment result of acceleration response for case 2 ($\zeta = 5000$).

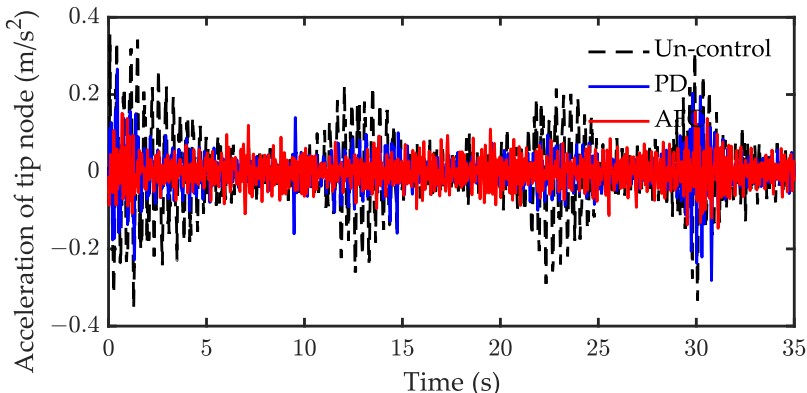

**Figure 24.** Experiment result of acceleration response for case 2 ($\zeta = 20{,}000$).

**Table 5.** Vibration suppression effect of the experimental system for case 2 ($\zeta = 2000$).

| Controller | Uncontrol | PD | AFC |
|:---:|:---:|:---:|:---:|
| RMS (m/s$^2$) | 0.0808 | 0.0413 | 0.0303 |
| Amplitude (m/s$^2$) | 0.3500 | 0.2819 | 0.2997 |

## 6. Conclusions

In this paper, a dynamic mathematical model for a flexible space manipulator was derived. To choose a suitable mode, the natural characteristics of the link under three different boundary conditions were compared to each other, and the dynamic response results of the simulation showed that the internal resonance behavior existed in the system. A computed torque controller was designed to track the angle of the joints. Furthermore, a model-independent adaptive fuzzy controller was proposed to suppress the elastic vibration. The simulations of the two cases were performed in a two-link flexible manipulator system, and the results show that the proposed control strategy had a good performance in rapidly tracking trajectory and effectively suppressing the flexible vibration. Experiments were also performed to verify the proposed control strategy and simulation results. Compared with the uncontrol, the RMS of the vibration response with AFC was attenuated by 78.3% and 81.3% in the two cases, respectively. In addition, to improve the accuracy of the mathematic model of the space manipulator system, the damping of the joint and link should be considered in future modeling studies, and the effect of the internal resonance phenomenon on the stability of the control system should be studied in further detail in the future.

**Author Contributions:** Conceptualization, C.F.; methodology, C.F.; validation, C.F. and M.S.; resources, S.N.; writing—original draft, C.F.; writing—review and editing, W.C.; supervision, W.C. and M.S.; project administration, W.C.; funding acquisition, M.S. All authors have read and agreed to the published version of the manuscript.

**Funding:** This work was supported by the National Natural Science Foundation of China under Grant No. 12102174, and the Research Fund of State Key Laboratory of Mechanics and Control for Aerospace Structures (Nanjing University of Aeronautics and Astronautics) (Grant No. MCMS-I-0122K01).

**Data Availability Statement:** Not applicable.

**Acknowledgments:** This work was supported by the National Natural Science Foundation of China under Grant No. 12102174, and the Research Fund of State Key Laboratory of Mechanics and Control for Aerospace Structures (Nanjing University of Aeronautics and Astronautics) (Grant No. MCMS-I-0122K01). The authors gratefully acknowledge these agencies for their support.

**Conflicts of Interest:** The authors declare no conflict of interest.

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
