# Peer review of "Trajectory Tracking and Adaptive Fuzzy Vibration Control of Multilink Space Manipulators with Experimental Validation"

_actuators, doi:10.3390/act12040138_

Round 1

Reviewer 1 Report

The proposed manuscript  titled Trajectory tracking and adaptive fuzzy vibration control of multi-link space manipulator with experimental validation dealt with  the problem of modeling and controlling of a space manipulator system with flexible joints and links. 

The authors  carried out some numerical simulation and experiment to verify the effectiveness of the proposed compound controller, and a comparison with proportional-derivative (PD) controller is given to highlight its superiority in suppressing residual vibration of the tip.

 The concluding remarks were supported by the data.

The paper is relevant and interesting  and Schematic of adaptive fuzzy controller was done with no errors.

Experimental results  ere done with merit.

A relevant point is  the  Experimental results show that the proposed control strategy  may  rapidly realize trajectory tracking and effectively suppress the flexible vibration

It seems to be original , according to other results from current literature, The sate of the  art may be done with a  number of  International authors.

 It deserves  publication after   revisions:

 The Numerical simulations  must show the efficiency of the control methods,  as well as the sensitivity of each control strategy to parametric errors.

Reviewer 2 Report

The topic of the paper is interesting. The authors can find below some questions and comments:

1) line 53: typo, correct Ricatti in Riccati

2) the introduction can be improved analyzing better the literature (important authors working on flexible arms are missing, e.g., De Luca, Malzhan, etc)

3) passage from eq 17 to eqs 37-39 is not clear. Could you please expand it? In particular, it is not clear to me how tau_s is related to the motor torques.. and what are tau and tau’ which have not been introduced

4) from eq 51, it seems that the output joint position is used for the control. Is it right? Or do you use the motor position? Also when you compared your control strategy with a simple PD, is the PD closed on the motor or output position? Clearly the performance and stability are different in the two cases. Please clarify. 

5) how is the link end displacement computed? Integration of accelerometer? Do you experience any drift and could it be a problem? In general, clarify better what are the sensors and actuators you are using and how you derive all the quantities used in the control laws. 

Reviewer 3 Report

The comments can be found in the uploaded file.

Round 2

Reviewer 1 Report

 In this proposed manuscript the authors dealt with  a dynamic mathematical model of flexible space manipulator.

To choose a suitable mode, the natural characteristics of the link under three  different boundary conditions are compared with each other, and the dynamic response  results of the simulation showed  the existence of  internal resonance in the  considered system.

Then,a   computed torque controller was post to track the angle of joints.

So, a model of  t adaptive fuzzy controller was  proposed to suppress the elastic vibration of the considered sysem.

The   simulations carried  out  of two cases were performed on a two-link flexible manipulator system, and  the  obtained results showed  that the proposed control strategy had a good performance in rapidly tracking trajectory and effectively suppressing the flexible vibration.

Experiments were  also  performed to verify the proposed control strategy and simulation results obtained.

 According to the obtained results with Un-control, the RMS of the vibration response with AFC attenuated by 78.3ï¼… and 81.3ï¼… 564 in the two cases, respectively.

In addition, the authors  improved  the accuracy of the mathematic  model of the space manipulator system, the damping of the joint and link can be future be  considered in the modeling, and the effect of the internal resonance phenomenon on the  stability of control system can be future studied in detail.

In summary, the paper  is well  written with no mistakes.

The problem proposed by the authors was    analyzed  with  relevant results  .

The paper posed figures with quality  and pertinent  discussions

I recommend publication as is

Reviewer 3 Report

The comments have been well addressed.